# DNA/RNA heteroduplex oligonucleotide technology for regulating lymphocytes in vivo

Masaki Ohyagi [1,2], Tetsuya Nagata [1,2✉], Kensuke Ihara [3], Kie Yoshida-Tanaka[1,2], Rieko Nishi [1,2], Haruka Miyata[1,2], Aya Abe[1,2], Yo Mabuchi [4], Chihiro Akazawa[4] & Takanori Yokota [1,2✉]

Manipulating lymphocyte functions with gene silencing approaches is promising for treating autoimmunity, inflammation, and cancer. Although oligonucleotide therapy has been proven to be successful in treating several conditions, efficient in vivo delivery of oligonucleotide to lymphocyte populations remains a challenge. Here, we demonstrate that intravenous injection of a heteroduplex oligonucleotide (HDO), comprised of an antisense oligonucleotide (ASO) and its complementary RNA conjugated to α-tocopherol, silences lymphocyte endogenous gene expression with higher potency, efficacy, and longer retention time than ASOs. Importantly, reduction of *Itga4* by HDO ameliorates symptoms in both adoptive transfer and active experimental autoimmune encephalomyelitis models. Our findings reveal the advantages of HDO with enhanced gene knockdown effect and different delivery mechanisms compared with ASO. Thus, regulation of lymphocyte functions by HDO is a potential therapeutic option for immune-mediated diseases.

[1] Department of Neurology and Neurological Science, Graduate School of Medical and Dental Sciences, Tokyo Medical and Dental University, Tokyo, Japan. [2] Center for Brain Integration Research, Tokyo Medical and Dental University, Tokyo, Japan. [3] Department of Bio-informational Pharmacology, Medical Research Institute, Tokyo Medical and Dental University, Tokyo, Japan. [4] Department of Biochemistry and Biophysics, Graduate School of Medical and Dental Sciences, Tokyo Medical and Dental University, Tokyo, Japan. ✉email: t-naga.nuro@tmd.ac.jp; tak-yokota.nuro@tmd.ac.jp

Lymphocytes are major cellular components of the adaptive immune system whose main functions are to recognize non-self-antigens and generate immune responses for eliminating specific pathogens. The two main types of lymphocytes are T cells and B cells. Both originate from stem cells in the bone marrow, but T cells mature in thymus and B cells mature in the bone marrow. The thymus and bone marrow constitute the primary lymphoid tissues that are the sites of lymphocyte generation and maturation, while the secondary lymphoid tissues, including lymph nodes and spleen, are responsible for maintaining mature naive lymphocyte and initiating an adaptive immune response through antigen presentation. Following maturation, lymphocytes recirculate through the blood and peripheral lymphoid organs where they survey for invading pathogens. However, inappropriate activation or inactivation of immune cells results in the development of chronic inflammatory diseases, cancer, or autoimmune diseases, such as multiple sclerosis (MS). Antibody blockade is a therapeutic strategy for these diseases, but these applications are restricted to primarily extracellular antigens, such as membrane receptors or secreted proteins[1,2]. Manipulating lymphocyte functions, by regulating intracellular molecules, aids not only in the understanding of the physiological and pathological roles of lymphocytes but also in the development of a molecular targeted therapy.

For specific knockdown of target genes, gapmer-type antisense oligonucleotides (ASOs) composed of DNA nucleotides flanked by artificially modified ribonucleotide monomers, such as locked nucleic acid (LNA) or 2'-O-methoxyethylribose modifications, are widely used as a research tool and therapeutic. ASOs are designed to specifically bind to a targeted RNA sequence, which can then modulate RNA function through promoting RNA cleavage and degradation or steric blocking mechanism. Most therapeutic ASO have phosphorothioate backbone modifications to enhance protein binding, stability in serum, and cellular uptake, presumably through ASO binding to serum or cell surface proteins[3]. Despite these chemical modifications, physical or chemical transfection methods such as electroporation or nucleofection are required for transduction of ASO into lymphocytes[4,5], typically leading to significantly decreased cell viability because of transient membrane permeabilization[6–8]. Viral-based vector systems mediate high transfection efficiencies, but severe immunogenicity and safety concerns have hindered their clinical applications. Other gene silencing techniques commonly applied to lymphocytes include nanoparticle- and aptamer-based interfering RNA delivery systems[9–11]. These studies support the development of targeted oligonucleotide delivery platforms for therapeutics of lymphocyte-based diseases.

We recently developed a "heteroduplex oligonucleotide" (HDO) approach that achieves highly efficient gene silencing in liver and brain microvascular endothelial cells in vivo[12,13]. HDO are composed of a parent ASO strand with gapmer structure, duplexed with complementary RNA (cRNA). HDOs conjugated with α-tocopherol (Toc-HDO), where the delivery ligand α-tocopherol is covalently conjugated to 5′-end of the cRNA, binds to serum lipoproteins in blood circulation, and are distributed via the α-tocopherol transport pathway[12,14]. In lymphocytes, α-tocopherol is physiologically essential to enhance their proliferation and form an effective immune synapse[15,16], thus becoming an optimal delivery ligand for lymphocytes. Given that ASOs are actively developed oligonucleotide agents[17], our HDO technology equipped with α-tocopherol has potential for gene silencing in lymphocytes in vivo.

Here we report that intravenously injected Toc-HDO can induce highly efficient and sustained knockdown of target molecules in mouse lymphocytes in peripheral blood and lymphoid tissues without any inflammatory reaction. Toc-HDO can ameliorate both adoptively transferred and active experimental autoimmune encephalomyelitis (EAE), a murine model of MS. We also show that the cellular uptake of Toc-HDO utilizes characteristic endocytic pathways. This study provides a productive modality for administrating a chemically synthesized oligonucleotide that can silence target genes in mouse lymphocytes with high efficacy and safety.

## Results

**Gene silencing efficacy of Toc-HDO in mouse lymphocytes in vivo.** In this study, we targeted *Integrin alpha 4* (*Itga4*; also known as *CD49d*), which plays key roles in leukocyte activation, trafficking, and signaling[18]. Two other target genes were *Metastasis-associated lung adenocarcinoma transcript 1* (*Malat1*; also known as *Neat2*) and *dystrophia myotonica-protein kinase* (*Dmpk*). The *Itga4* ASO sequence was originally designed for this study (Supplementary Fig. 1a and Supplementary Table 1). Specific ASO sequences for *Malat1* and *Dmpk* have high specificity, efficacy, biological stability, and safety in previous studies[19–21] (Supplementary Fig. 1b). We then annealed an α-tocopherol-bound 16-mer cRNA to each ASO sequence (Fig. 1).

First, we evaluate the gene silencing effect of ASO, Toc-HDO, and directly conjugated α-tocopherol to 5'-end of ASO (Toc-ASO) targeting *Malat1* in primary T cells in vitro. Although the conventional ASO induced target gene reduction, Toc-HDO induced more efficient gene knockdown. The direct conjugation of α-tocopherol drastically decreased the gene silencing effect even at high doses (Supplementary Fig. 1c). To examine the in vivo efficacy of Toc-HDO-mediated endogenous gene targeting, mice were intravenously injected with *Itga4*-targeting Toc-HDO or ASO at doses corresponding to 50 mg/kg of the parent ASO. After 72 h, Toc-HDO significantly reduced *Itga4* mRNA expression compared with an equivalent dose of ASO in peripheral blood, splenic, lymph node, and thymic lymphocytes and bone marrow cells (Fig. 2a). Similar findings were observed using the 16-mer Toc-HDO targeting *Malat1* RNA and *Dmpk* mRNA at the same ASO doses (Fig. 2b and Supplementary

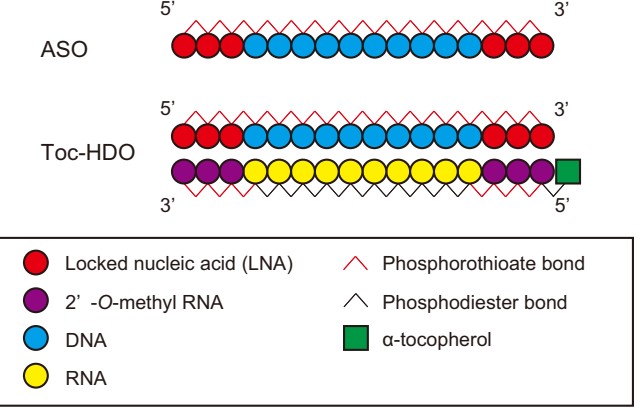

**Fig. 1 Design of ASO and Toc-HDO.** Schematic illustrations of ASO and Toc-HDO construction. HDO is composed of a gapmer ASO, duplexed with a complementary RNA (cRNA) conjugated to delivery ligands, in which cRNA is cleaved by endogenous RNase H in the nucleus activating ASO. Gapmer ASOs have a central gap region of DNA flanked by modified nucleotides that enhance affinity for cRNA, such as locked nucleic acid (LNA) or 2'-O-methyl RNA and whole nucleotides of ASO strand are protected from nucleases by phosphorothioate (PS) modification. In the cRNA strand, the center portion hybridized to the gap portion of ASO remains unmodified for recognition of RNase H, and terminal nucleotides hybridized to both wing portions of ASO are protected from exonucleases by both PS modification and 2'-O-methyl RNA.

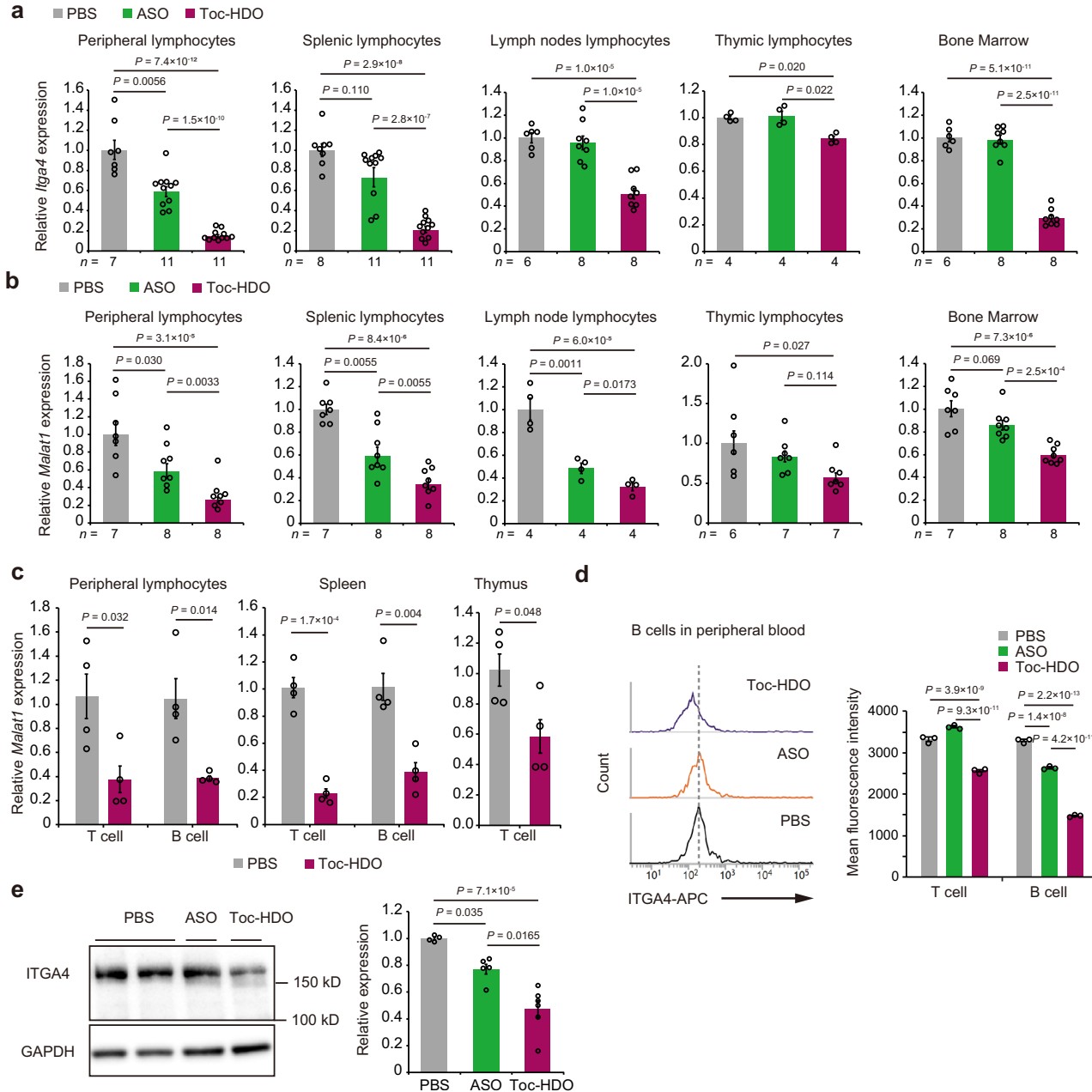

**Fig. 2 Gene silencing by intravenous administration of Toc-HDO targeting endogenous genes in mouse lymphocytes in vivo. a, b** Target mRNA levels measured by quantitative RT-PCR in mouse lymphocytes from the indicated tissues 72 h after intravenous injection of 50 mg/kg Toc-HDO, ASO, or PBS alone. **a** *Itga4*. **b** *Malat1*. **c** Quantitative RT-PCR analyses of *Malat1* RNA in mouse T cells and B cells from the indicated tissues 72 h after intravenous injection of 50 mg/kg Toc-HDO, ASO, or PBS alone ($n = 4$ for each group). **d, e** ITGA4 protein expression determined by flow cytometry in peripheral blood lymphocytes (**d**; $n = 3$ for each group) and by western blot in splenic lymphocytes (**e**; PBS, $n = 4$; ASO, $n = 5$; Toc-HDO, $n = 6$) 5 days after intravenous administration of 50 mg/kg Toc-HDO, ASO, or PBS alone. Quantitative RT-PCR data were normalized to *Gapdh* mRNA levels. Data are expressed as mean ± s.e.m. and represent at least two independent experiments. *P* values were calculated using one-way ANOVA with Holm's post-test. Source data are provided as a Source data file.

Fig. 2a). We also evaluated the in vivo gene silencing efficacy of Toc-ASO, but all mouse died within 2 days after intravenous injection of Toc-ASO at corresponding doses to 50 mg/kg of the ASO, indicating high toxicity with direct conjugation of α-tocopherol to ASO in vivo. These results suggest that systemic administration of ASO has a mild gene silencing effect on lymphocytes and direct conjugation of α-tocopherol to ASO prevents the ASO activity and leads to a fatal outcome in vivo, whereas Toc-HDO reduces target gene expression more efficiently than ASO in lymphocytes in vivo, especially peripheral

blood lymphocytes. We further checked whether off-target effects occur between each of target sequences and the expression of *Epsin 2* (*Epn2*) mRNA, which is 14 bp matched with *Itga4*-targeted ASO sequence; however, each of Toc-HDO and ASO targeting *Itga4* and *Malat1* was specific for the target gene (Supplementary Figs. 1d and 2b).

To further investigate the in vivo efficacy of Toc-HDO in T or B cells in peripheral blood, spleen, and thymus, we performed fluorescence-activated cell sorting 72 h after intravenous injection of *Malat1* RNA-targeting Toc-HDO using anti-CD3 antibody for

**Table 1 Biochemical analysis of experimental autoimmune encephalomyelitis mouse serum after administration of PBS, ASO, or Toc-HDO.**

|  | AST (U/l) | ALT (U/l) | T-Bil (mg/dl) | BUN (mg/dl) | Cre (mg/dl) |
|---|---|---|---|---|---|
| PBS | 57 ± 1.5 | 19 ± 1.9 | 0.027 ± 0.003 | 35.0 ± 4.4 | 0.14 ± 0.01 |
| ASO | 67 ± 5.5 | 18 ± 1.4 | 0.040 ± 0.005 | 24.4 ± 2.7 | 0.12 ± 0.01 |
| Toc-HDO | 55 ± 1.9 | 15 ± 0.4 | 0.030 ± 0.001 | 26.2 ± 0.6 | 0.12 ± 0.01 |

Data are presented as mean ± s.e.m. P values were determined using one-way ANOVA with Holm's post-test. No significant differences were detected.
*ALT* alanine aminotransferase, *ASO* antisense oligonucleotide, *AST* aspartate aminotransferase, *BUN* blood urea nitrogen, *Cre* creatinine, *T-Bil* total bilirubin, *Toc-HDO* α-tocopherol-conjugated heteroduplex oligonucleotide.

T cells and anti-CD45R/B220 for B cells. Toc-HDO silences target gene expression in both T and B cells (Fig. 2c). Furthermore, ITGA4 protein expression on the T and B cell membranes was reduced 5 days after a Toc-HDO single injection (Fig. 2d). Similar data were obtained from splenic lymphocytes by western blot and flow cytometric analysis (Fig. 2e and Supplementary Fig. 2c).

Additionally, we evaluated the adverse effects of Toc-HDO. Serum biochemistry analyses did not show any abnormalities with Toc-HDO doses up to 50 mg/kg (Table 1). Then, we examined pro-inflammatory cytokine expression in lymphocytes. *Dmpk*-targeting ASOs induced *Tumor necrosis factor-α (TNF-α)*, whereas Toc-HDO had no significant effect (Supplementary Fig. 2d). Similar tendency was seen for oligonucleotides targeting *Itga4*, but the result was not statistically significant (Supplementary Fig. 2e). These results suggest that Toc-HDO is superior to ASO with regard to immune stimulation, which poses a difficult problem in the clinical application of nucleic acid therapy.

**Pharmacokinetics, pharmacodynamics, and biodistribution of Toc-HDO in vivo.** We next tested the in vivo dose-dependent effects of Toc-HDO and ASO. After injection of Toc-HDO targeted to *Itga4* or *Dmpk* mRNA, target mRNA was dose-dependently reduced (Fig. 3a and Supplementary Fig. 2f). Injection of ASO targeting *Itga4* did not reduce target mRNA in similar dose ranges. The 50% effective dose ($ED_{50}$) of Toc-HDO targeting *Itga4* in peripheral and splenic lymphocytes was 21.30 and 18.44 mg/kg, respectively. Furthermore, the 80% effective dose ($ED_{80}$) of Toc-HDO targeted to *Dmpk* in splenic lymphocytes was 13.57 mg/kg, which was 1.6-fold more potent than the ASO, whose $ED_{80}$ was 22.86 mg/kg (Supplementary Fig. 2f). In thymic lymphocytes, Toc-HDO was more efficient (90% inhibition at 25 mg/kg) compared to ASO (75% inhibition at 25 mg/kg). The corresponding $ED_{50}$ was 16.98 mg/kg, which was 1.4-fold more potent than the ASO, whose $ED_{50}$ was 22.07 mg/kg (Supplementary Fig. 2f).

Next, we tested the gene silencing duration in lymphocytes after a single dose of Toc-HDO or ASO. After injection, Toc-HDO-mediated target RNA reduction in peripheral blood lymphocytes was maximal on day 3 and lasted for >1 month (Fig. 3b). Surprisingly, the Toc-HDO effect compared to ASO also lasted for >1 month. Reduction of the target RNA in splenic and thymic lymphocytes was also maximal on day 3 and lasted 14 days (Fig. 3b).

We further examined the persistence of Toc-HDO in lymphocytes to explore the mechanism of longitudinal gene silencing after a single dose of Toc-HDO. We injected 20 mg/kg ASO oligonucleotides with 5' Alexa Fluor 647 labels into the tail vein and examined their presence in plasma and lymphocytes 1, 3, 6, 12, and 24 h after injection. Significant retention of Alexa

Fluor 647-labeled oligonucleotides was observed in plasma at 12 h and lymphocytes treated with Toc-HDO even at 24 h after injection (Fig. 3c). Flow cytometric analysis revealed that both T cells and B cells in peripheral blood and spleen 6 h after injection efficiently internalized fluorescence-labeled Toc-HDO compared with ASO (Fig. 3d). To examine the biodistribution of Toc-HDO, we further injected 20 mg/kg 5' Cy5-labeled ASO oligonucleotides into the tail vein and examined the presence and localization of peripheral blood lymphocytes 1 and 6 h later. Peripheral blood lymphocytes had taken up Cy5-labeled Toc-HDO in some lymphocytes 1 h after injection (Fig. 3e(i)), but Cy5-labeled ASO was not observed (Fig. 3e(iii)). Cy5-labeled Toc-HDO was diffusely distributed in lymphocyte nuclei 6 h after injection (Fig. 3e(ii)), whereas ASO was observed in the nucleus of some lymphocytes (Fig. 3e(iv), (v)). Collectively, intravenously administered Toc-HDO induces continuous reduction of target gene expression through increased blood retention, is more quickly taken up and localized to peripheral blood lymphocyte nuclei, and persists longer than ASO at cellular levels.

**Toc-HDO internalization into lymphocytes through a pathway distinct from that used by ASOs.** We next analyzed the efficacy of gymnotically delivered Toc-HDO in the EL4 murine cultured T cell line. First, EL4 cells were incubated with 10 nM to 1 μM Toc-HDO for 24 h without any transfection reagents. In these cells, Toc-HDO induced significant gene silencing in a dose-dependent manner (Fig. 4a). EL4 cells were also incubated with Toc-HDO for 24 h or 4 h followed by 20 h culture. RNA was 50% decreased after 4 h incubation, and a significant decrease was evident after 24 h incubation (Fig. 4b). Because 4 h Toc-HDO incubation followed by 20 h culture results in consistent target mRNA knockdown by about 50%, we used this assay format in the following study.

There are many possible pathways that can internalize small molecules, including ASO or oligonucleotides, in mammalian cells[22,23]. These include pinocytosis, phagocytosis, clathrin-mediated endocytosis, and caveolae-mediated endocytosis. We previously found that Toc-HDO binds to serum lipoproteins, especially high-density lipoproteins (HDLs), and is delivered via the α-tocopherol pathway to hepatocytes[12]. We hypothesized that Toc-HDO enters mouse lymphocytes through clathrin-mediated endocytosis, because HDL-bound α-tocopherol is efficiently internalized through clathrin-mediated endocytosis via scavenger receptor type I or low density lipoprotein (LDL) receptors[24]. To investigate the delivery mechanism of Toc-HDO into mouse lymphocytes, we treated mouse cultured lymphocytes separately with cool stimulation (Supplementary Fig. 3a) or specific endocytosis pathway inhibitors including amiloride[25–28], chlorpromazine[26,29,30], dynasore[31], filipin[29,32,33], and cytochalasin D[34,35], which block pinocytosis, clathrin-mediated endocytosis, dynamin, caveolae pathways, and phagocytosis, respectively. Pretreatment of EL4 with increasing concentrations of amiloride 1 h prior to Toc-HDO or ASO incubation showed dose-dependent inhibition of gene silencing efficacy (Fig. 4c). Similar results were observed in ASO-mediated gene silencing with dynasore treatment. However, about 30% Toc-HDO-mediated target gene reduction occurred, even with a large dose of dynasore (Fig. 4d). The higher concentrations of dynasore were found to be cytotoxic (Supplementary Fig. 3b). Inhibitory concentrations of chlorpromazine, filipin, or cytochalasin D had no significant effect on either Toc-HDO or ASO efficacy (Fig. 4e, f and Supplementary Fig. 3c). To further validate the mechanisms that internalize Toc-HDO molecules in EL4 cells, we measured Alexa Fluor 647-conjugated oligonucleotides in the presence or absence of amiloride or dynasore by flow cytometry and fluorescence

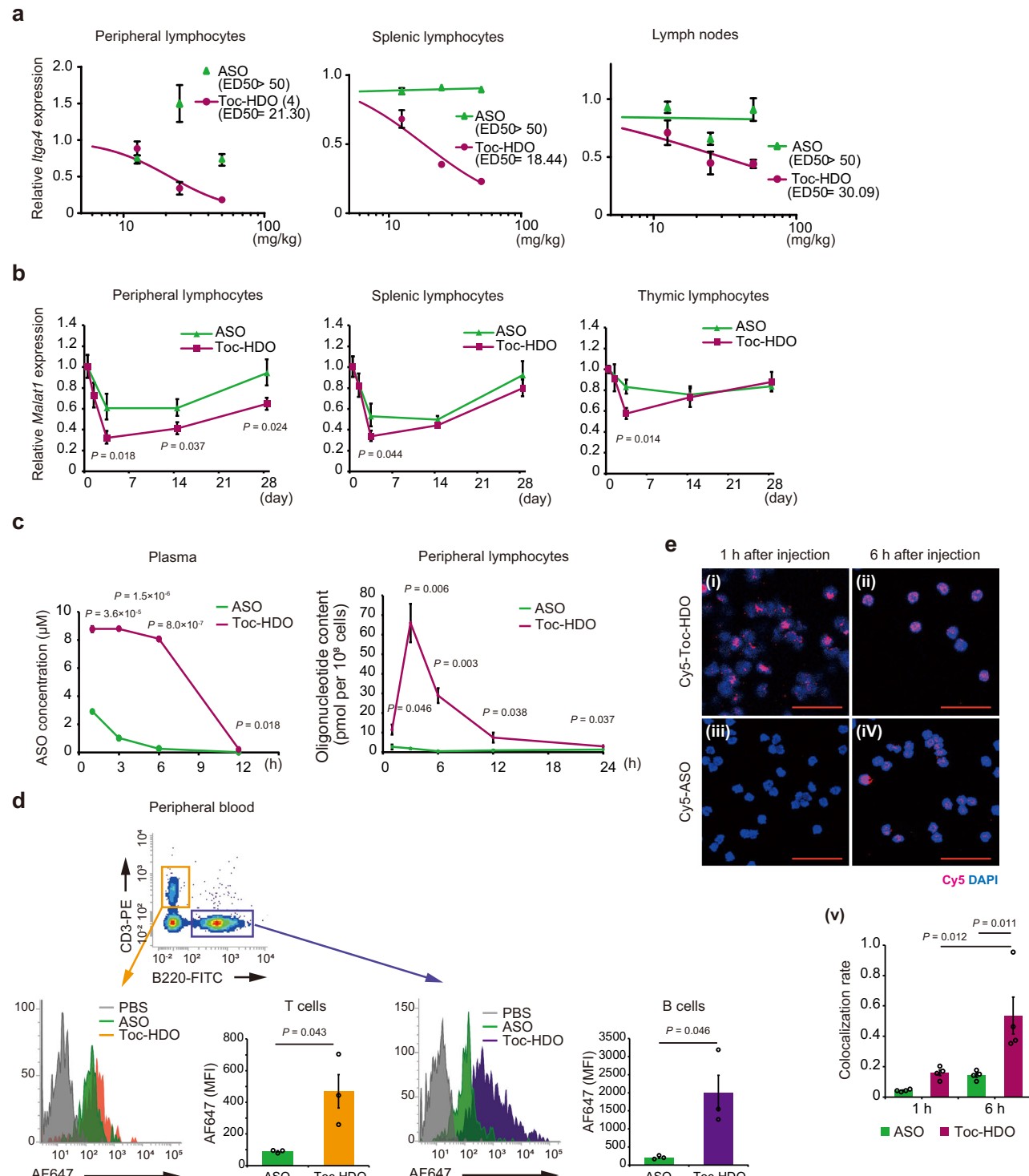

**Fig. 3 Intravenously administered Toc-HDO results in dose-dependent gene silencing, efficient cellular uptake, and high retention in mouse lymphocytes. a** Dose-dependent reduction of *Itga4* mRNA levels in mouse lymphocytes from indicated tissues 72 h after intravenous injection of Toc-HDO or ASO ($n = 4$ for each group). **b** Time course of *Malat1* RNA levels in mouse lymphocytes after injection of 50 mg/kg Toc-HDO or ASO over 4 weeks ($n = 4$ for each group). **c** Time course of oligonucleotide concentration in plasma and peripheral blood lymphocytes after injection of 20 mg/kg Alexa Fluor 647 (AF647)-labeled Toc-HDO or ASO over 24 h. Oligonucleotide concentration was calculated by fluorescence intensity ($n = 3$ for each group). **d** Representative histogram and quantitative data of AF647-labeled Toc-HDO or ASO internalized by T cell and B cell from peripheral blood 6 h after intravenous injection ($n = 3$ for each group). MFI mean fluorescence intensity. **e** Confocal laser scanning microscopic images of mouse peripheral blood lymphocytes at 1 h (**i**, **iii**) or 6 h (**ii**, **iv**) after intravenous administration of 20 mg/kg Cy5-labeled Toc-HDO (upper) or ASO (lower). Sections were stained with DAPI. Red, Cy5; blue, DAPI. Scale bars, 50 μm. Images shown are representative of two experiments. Colocalization of Cy5-labeled Toc-HDO or ASO with DAPI at the indicated time point were measured by percentage of area occupied by Cy5 in DAPI-positive areas (**v**; $n = 4$ for each group). Data are expressed as mean ± s.e.m. Data shown are representative of two experiments. *P* values were calculated using one-way ANOVA with Holm's post-test (**b**, **e**(**v**)) or two-sided Student's two-tailed *t* test (**c**, **d**). Source data are provided as a Source data file.

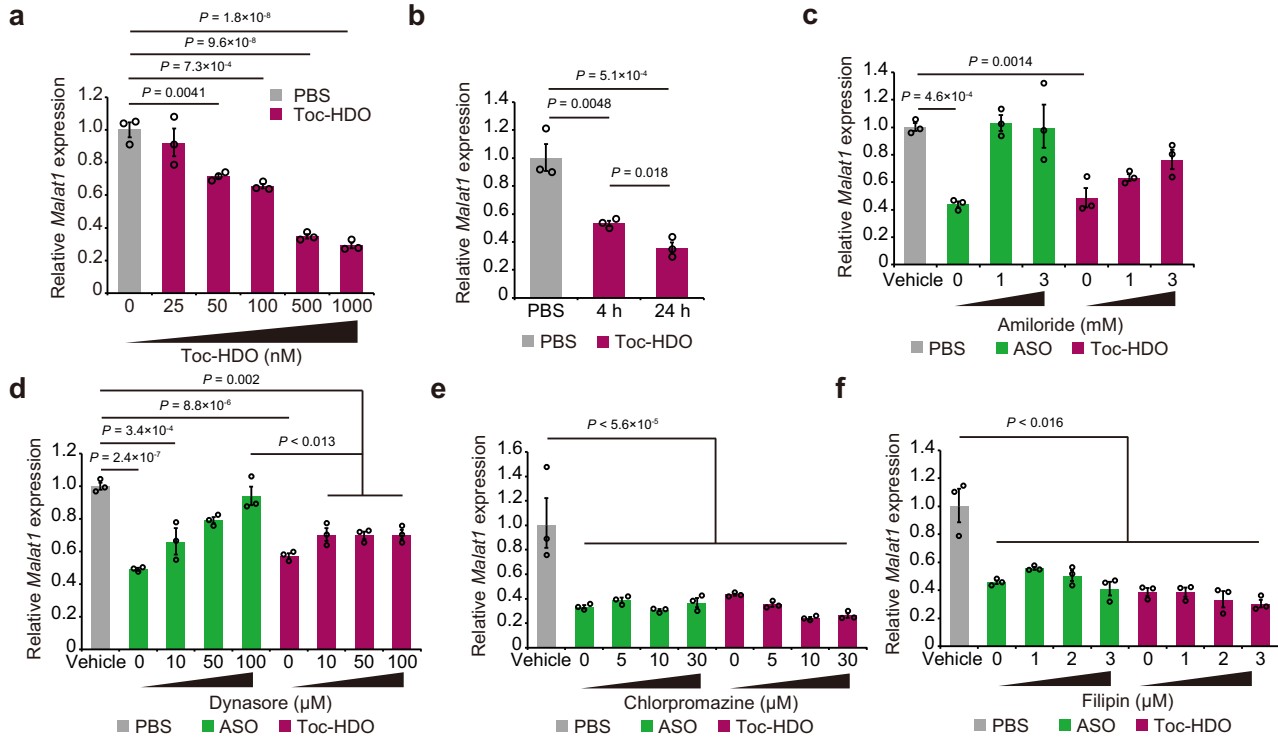

**Fig. 4 Endocytosis inhibitors reduce Toc-HDO cellular internalization in mouse T cells. a** Quantitative RT-PCR analyses of *Malat1* RNA levels in EL4 cells, which were incubated with Toc-HDO for 24 h without any transfection reagents. **b** *Malat1* RNA levels in EL4 cells after incubation with 500 nM Toc-HDO. EL4 cells were incubated with 500 nM Toc-HDO for 24 or 4 h followed by 20 h culture. **c–f** Quantitative RT-PCR analyses of *Malat1* RNA levels in EL4 cells treated with amiloride (**c**), dynasore (**d**), chlorpromazine (**e**), or filipin (**f**), followed by treatment with Toc-HDO or ASO. EL4 cells were incubated with each endocytosis inhibitor for 1 h and then treated with 500 nM Toc-HDO or ASO for 4 h, followed by 20 h culture. Data are normalized to *Gapdh* mRNA levels and represent at least two independent experiments ($n = 3$ for each group). Data are expressed as mean ± s.e.m. *P* values were calculated using one-way ANOVA with Holm's post-test. Source data are provided as a Source data file.

microscope. Amiloride and dynasore both abolished ASO and Toc-HDO internalization (Supplementary Fig. 3d–f). Taken together, ASO uptake depends on macropinocytosis, as reported previously[8], and on dynamin-dependent uptake pathways, but not on clathrin- or caveolae-dependent pathways. Toc-HDO is also internalized via macropinocytosis and dynamin-dependent pathways with varying degrees. However, Toc-HDO differs from ASO in that Toc-HDO can utilize the dynamin-independent pathway as an alternative internalization pathway (Supplementary Fig. 4). These mechanisms were supported by the findings of cellular uptake analyses using fluorescence-conjugated oligonucleotides (Fig. 3d, e). Flow cytometry and fluorescent microscopy revealed that, although lymphocytes take up ASO, their cellular uptake of Toc-HDO is more quickly and efficiently, indicating that α-tocopherol conjugation induces a passive targeting effect that facilitates ligand binding to serum proteins in order to decrease clearance and increase circulation time (Fig. 3c), rather than an active targeting effect that a ligand binds to a specific receptor expressed on the target cells. HDO conjugated with α-tocopherol allows prolonged blood retention time and efficient internalization through several uptake pathways at cellular levels.

For investigation into gene silencing effect of Toc-HDO in human lymphocytes, we evaluated target gene knockdown by Toc-HDO in human cultured lymphocyte. We targeted *Stat3*, which acts as a transcription activator downstream of many cytokines and growth factors receptors expressed on lymphocytes, because specific ASO for *Stat3* has high efficacy and safety in clinical studies[36]. Toc-HDO targeting *Stat3* demonstrated significant target gene knockdown at a high dose in Jurkat cells compared to ASO (Supplementary Fig. 5a). At protein levels,

Toc-HDO reduced STAT3 expression (Supplementary Fig. 5b). These findings support clinical application of Toc-HDO in regulating human lymphocyte functions.

**Toc-HDO-mediated suppression of *Itga4* prevents EAE and graft versus host disease (GVHD).** We examined the potency of Toc-HDO as a new therapeutic platform in autoimmune diseases. ITGA4, expressed on the surface of inflammatory lymphocytes, plays a critical role in adhesion of circulating lymphocytes to the vascular endothelium and infiltration into the central nervous system (CNS). The migration of autoreactive T cells into the CNS has been identified as a crucial step in the formation of MS lesions; thus, a blocking antibody is one of the most effective therapy for MS patients[37]. In this, we aimed Toc-HDO-mediated knockdown of *Itga4* to treat EAE. We beforehand confirmed that *Itga4*-targeting ASO and Toc-HDO have no effect on cell viability and dysfunctional cytokine production (Supplementary Fig. 6a, b). We prophylactically treated active EAE mice with 20 mg/kg Toc-HDO targeting *Itga4* by intravenous injection twice weekly for a total of five injections. Toc-HDO significantly delayed EAE onset compared to ASO or phosphate-buffered saline (PBS) alone (Fig. 5a). Immunohistochemical (IHC) analysis also revealed a marked reduction in CD4+ T cell infiltration (Fig. 5b), spinal cord demyelination, and inflammatory cell infiltration after Toc-HDO treatment (Fig. 5c, d). *Glial fibrillary acidic protein* (*Gfap*), *Induction of brown adipocytes 1* (*Iba1*), and *Interleukin-1β* (*Il-1β*) expression in the spinal cord of Toc-HDO-treated mice was markedly downregulated, indicating that Toc-HDO reduces reactive gliosis and inflammation (Fig. 5e). To evaluate the

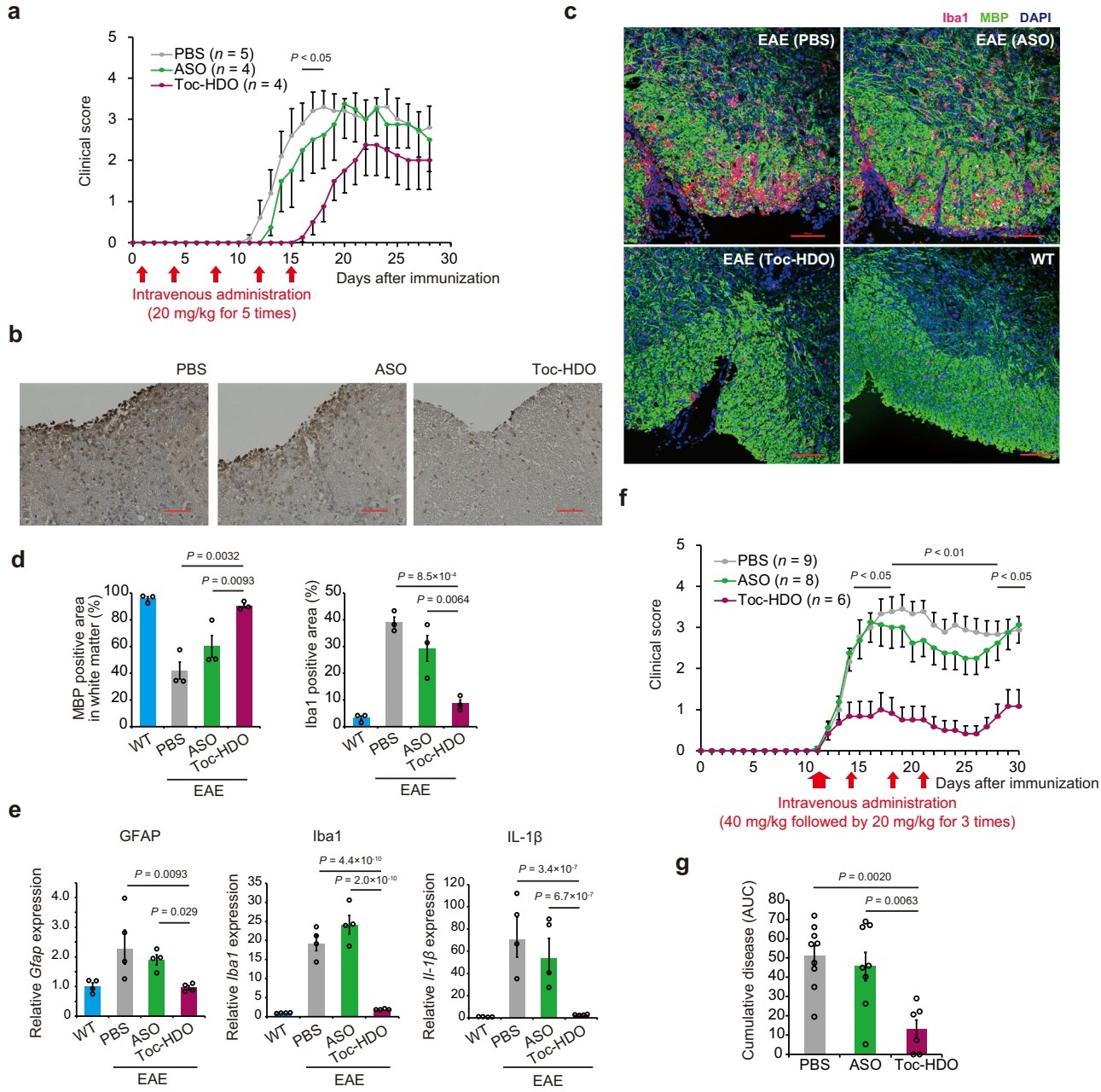

**Fig. 5 Toc-HDO targeting *Itga4* ameliorates clinical EAE symptoms and suppresses spinal cord demyelination and immune cell infiltration. a** Clinical EAE scores of mice after prophylactic treatment with Toc-HDO, ASO, or PBS alone twice weekly for a total of five injections after immunization. **b** Immunohistochemical staining of CD4$^+$ T cells in mouse spinal cord sections after treatment with Toc-HDO, ASO, or PBS alone. Scale bars, 50 μm. Images shown are representative of two experiments. **c** Immunohistochemical staining for myelin basic protein (MBP) and Iba1 in mouse spinal cord sections after treatment with Toc-HDO, ASO, or PBS alone, and wild-type (WT) mice. Scale bars, 100 μm. Images shown are representative of two experiments. **d** Percentage of MBP-positive area in spinal cord white matter, and the percentage of Iba1-positive area in spinal cord ($n = 3$ for each group). **e** Quantitative RT-PCR analysis of *Gfap*, *Iba1*, and *Il-1β* mRNA in lumbar spinal cord tissue of EAE mice after treatment with Toc-HDO, ASO, or PBS alone, and WT mice ($n = 4$ for each group). **f** Clinical EAE scores of mice therapeutically treated with Toc-HDO, ASO, or PBS alone after EAE symptoms onset. **g** The severity of EAE disease evaluated by the area under the curve (AUC). Data from PBS ($n = 9$), ASO ($n = 8$), and Toc-HDO ($n = 6$). Data are expressed as mean ± s.e.m. and represent at least two independent experiments. *P* values were calculated using one-way ANOVA with Holm's post-test. Source data are provided as a Source data file.

therapeutic effect of Toc-HDO on disease progression after onset, we further intravenously administered 40 mg/kg Toc-HDO targeting *Itga4* after EAE symptoms onset, followed by 20 mg/kg intravenous injection twice weekly for a total of three injections. Therapeutic treatment by Toc-HDO remarkably ameliorated EAE severity compared with ASO or PBS alone (Fig. 5f). The area

under the curve for clinical EAE score in mice treated with Toc-HDO and ASO reduced by 77.1 and 10.9%, respectively (Fig. 5g).

We then investigated ex vivo Toc-HDO treatment efficacy in an adoptive transfer EAE model. Because adoptive transfer of MOG$_{35-55}$-primed T cells into wild-type mice induces EAE, we examined whether Toc-HDO exposure during ex vivo culture of MOG$_{35-55}$-primed T cells was sufficient to inhibit EAE

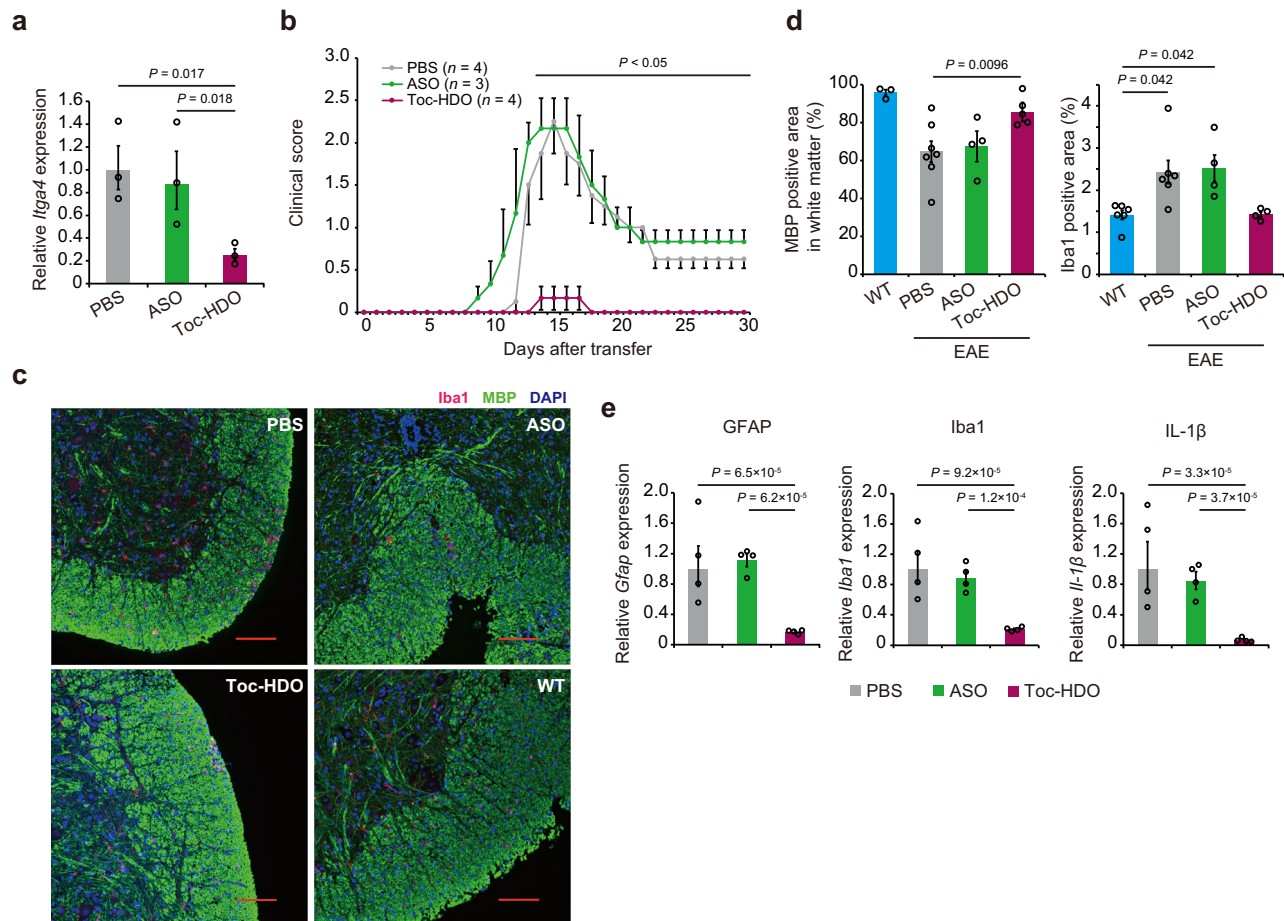

**Fig. 6 Adoptive transfer of primary T cells treated with Toc-HDO ex vivo suppresses experimental autoimmune encephalomyelitis. a** Quantitative RT-PCR analyses of *Itga4* mRNA levels in MOG$_{35-55}$ peptide-primed T cells incubated with 1 μM Toc-HDO, ASO, or PBS alone for 72 h without any transfection reagents ($n = 3$ for each group). **b** Clinical EAE scores of mice after transfer of MOG$_{35-55}$-primed T cells treated with Toc-HDO, ASO or PBS alone. **c** Immunohistochemical staining for myelin basic protein (MBP) and Iba1 in mouse spinal cord sections after treatment with Toc-HDO, ASO, or PBS alone and wild-type (WT) mice. Scale bars, 100 μm. Images shown are representative of two experiments. **d** Percentage of MBP-positive area in the spinal cord white matter, and percentage of Iba1-postive area in the spinal cord. Data from WT ($n = 3$), PBS ($n = 7$), ASO ($n = 4$), and Toc-HDO ($n = 5$) for MBP-positive area and WT ($n = 6$), PBS ($n = 6$), ASO ($n = 4$), and Toc-HDO ($n = 4$) for Iba1-positive area **e** Quantitative RT-PCR analysis of *Gfap*, *Iba1*, and *Il-1β* mRNA in lumbar spinal cord tissue of EAE mice after adoptive transfer of MOG$_{35-55}$-primed T cells treated with Toc-HDO, ASO, or PBS alone ($n = 4$ for each group). Quantitative RT-PCR data are normalized to *Gapdh* mRNA levels. Data are expressed as mean ± s.e.m. Data represent at least two independent experiments. *P* values were calculated using one-way ANOVA with Holm's post-test. Source data are provided as a Source data file.

development. Lymphatic or splenic T cells from MOG$_{35-55}$-primed mice were stimulated with MOG$_{35-55}$ peptide in the presence of Toc-HDO, ASO, or PBS (Fig. 6a). Then these MOG$_{35-55}$-primed T cells were adoptively transferred to recipient wild-type mice. The mice that received Toc-HDO-treated MOG$_{35-55}$-primed T cells exhibited significant reduction of clinical symptoms compared to mice that received non-treated or ASO-treated MOG$_{35-55}$-primed T cells (Fig. 6b). IHC and gene expression analyses also showed a reduction in spinal cord demyelination, inflammation, and reactive gliosis after Toc-HDO-treated MOG$_{35-55}$-primed T cell transfer (Fig. 6c–e).

Finally, we also examined HDO-mediated *Itga4* regulation in acute GVHD model. GVHD remains a major cause of morbidity and mortality in allogeneic hematopoietic cell transplantation. A fully major histocompatibility complex (MHC)-mismatched transplantation with T cell-depleted bone marrow cells (TCD-BM) along with spleen T cells results in GVHD, but not transplantation with only TCD-BM. In this model, the crucial role of Itga4 in the pathology of GVHD[38,39] offers inhibition of Itga4 in allo-reactive T cell migration to peripheral tissues as a therapeutic approach for GVHD. We initially isolated spleen

T cells from B6 mice intravenously injected with PBS, ASO, or Toc-HDO targeting *Itga4*. Then we intravenously injected these spleen T cells along with TCD-BM isolated from non-treated B6 mice into lethally irradiated allogeneic BALB/c recipient mice. Mice transplanted with only TCD-BM did not develop GVHD as previously reported[39,40] (Supplementary Fig. 6c). In contrast, daily monitoring revealed significant lower GVHD clinical score in mice injected with Toc-HDO-treated T cells than in those injected with PBS- and ASO-treated T cells (Supplementary Fig. 6d). Survival rate tended to be higher in the Toc-HDO-treated T cell group compared with the PBS- and ASO-treated T cell group but not significant (Supplementary Fig. 6c). Taken together, modulation of lymphocyte function by Toc-HDO technology is a potent therapeutic strategy for autoimmune diseases, inflammation, and transplantation.

## Discussion
Efficient gene silencing in lymphocytes using therapeutic oligo-nucleotides remains challenging, since these cells are highly resistant to transfection owing to their thin cell membrane, limited endocytosis, and lower cell surface protein content[41,42].

Lymphocytes also express a variety of nucleic acid-sensing machineries such as Toll-like receptors, RIG-1 like receptor, and cGAS/STING pathways[43–45], which can induce inflammatory response, apoptosis, and necrosis after transfection of oligonucleotides[6,46]. Furthermore, lymphocytes are distributed throughout the body through blood and lymphatic vessels and are often located in deep tissues[42,47]. Therefore, for in vivo delivery to lymphocytes, conventional RNA-based methods still show insufficient delivery, poor stability, and poor bioavailability along with their large molecular weight, negative charge, and hydrophilicity. For example, chemically modified small interfering RNA (siRNA) can be transduced using viral vehicles, transfection reagents, or electroporation in vitro, but these methods may cause significant cell damage or severe immune responses[4,7,8,47]. For efficient in vivo transfection of lymphocytes, other approaches such as lipid nanoparticles[9,48], aptamer[11,49], or antibody fragment fusion protein conjugates[50–52] have been applied. Self-deliverable and chemically modified ASO has achieved highly efficient and specific target gene suppression[17,53]. Indeed, ASO itself has been reported to exhibit gene suppression in mouse or monkey lymphoid tissues or lymphocytes from EAE lymph nodes[19,54]. However, a limited number of studies support this conclusion, and in vivo gene silencing effects of ASO targeted to lymphocytes remain insufficient and are not optimized for clinical translation[55]. Herein we have demonstrated the robust gene silencing of three different lymphocyte endogenous genes by Toc-HDO with the improved potency ($ED_{50}$) in most lymphoid tissues evaluated, especially circulating lymphocytes in vivo. Improved efficacy (maximal reduction) was also observed in mice treated with 50 mg/kg Toc-HDO, especially Toc-HDO targeting *Itga4*. Since circulating lymphocytes and lymphocytes in other lymphoid tissues play a crucial role in autoimmune disease, inflammation, and cancer immunology[56,57], our technology is a more therapeutically applicable approach not previously available by a conventional ASO.

In this study, we used α-tocopherol conjugation as a delivery ligand for lymphocytes because α-tocopherol is physiologically essential to enhance their proliferation and cytokine production[16,58], form an effective immune synapse[15], and prevents exhaustion and apoptosis[59,60]. Further, α-tocopherol is innocuous even at high doses[61]. HDO-based technology successfully results in target gene knockdown in liver and brain endothelial cells[12,13], not necessarily specific to lymphocytes. Conjugation with α-tocopherol allows HDO to bind to serum lipoprotein such as HDL and LDL, resulting in an efficient delivery along with physiological pathway of α-tocopherol[12]. Unconjugated HDO as well as ASO is small enough to be rapidly filtered out of blood by the kidney and excreted in urine, so binding to serum proteins enhanced by ligand conjugation maintains HDO in circulation long enough to distribute broadly to peripheral tissues. Efficient cellular uptake of Toc-HDO was attributed to multiple roles of α-tocopherol conjugation, including high retention in blood and escape from renal excretion with the passive targeting effect, and efficient internalization through multiple cellular uptake mechanisms. However, we have to pay attention to the possibility of unexpected adverse effects by α-tocopherol conjugation. Our platform, meanwhile, can equip various candidate ligands with cRNA without interfering with ASO access to a target RNA, thus allowing highly efficient delivery to targeted tissues or cells at lower oligonucleotide doses with lower adverse effects. Because lymphocytes can be classified into various subsets that uniquely express surface molecules, using cell surface markers as targeting moieties may offer an advantage for lymphocyte subset-specific delivery and gene manipulation in vivo. For example, aptamer for CD137, a major immune stimulatory receptor transiently expressed on activated CD8[+] T cells, was conjugated to siRNA targeting the intrinsic gene in circulating CD8[+] T cells in vivo[62]. Similarly, siRNA conjugated to CD4 aptamers facilitated the CD4[+] T cell-specific target gene silencing[11]. Cell surface molecules that are modified or upregulated in activated lymphocytes, such as LFA-1 and CD40L, may enable activation-dependent oligonucleotide delivery[51,63]. Further investigation into a suitable or optimized ligand to target specific subsets of lymphocytes may provide more specific HDO-based experimental tools for basic research and potential immunotherapeutic applications.

Previous studies have shown that cell surface proteins are involved in ASO cellular uptake, including cell surface receptors such as integrins, G-protein-coupled receptors, scavenger receptors, and epidermal growth factor receptor[64–67]. In contrast, ASO internalization into primary T cells uses a macropinocytosis pathway[8]. Several pathways described above have been productive, which means that uptake results in pharmacological knockdown effects, whereas other pathways may be nonproductive[22,68]. Thus, we explored the Toc-HDO endocytic pathways by evaluating knockdown effects in the presence of various endocytosis inhibitors. Our results showed that Toc-HDO enters mouse lymphocytes by macropinocytosis and clathrin-independent dynamin-dependent pathways, which are common to ASO. Further, Toc-HDO is also uniquely uptaken by a clathrin- and dynamin-independent pathway. These data suggest that Toc-HDO uptake via multiple pathways may contribute its rapid cellular uptake and high efficacy.

Here we found that intravenous administration of Toc-HDO efficiently downregulates endogenous gene expression in mouse lymphocytes with more efficient cellular uptake and longer retention time than the parent ASO. Furthermore, our technology suppressed disease development and progression of both active and adoptively transferred EAE in mice. Toc-HDO treatment for EAE inhibited inflammatory cell infiltration into the spinal cord, decreased inflammatory molecule expression, and prevented demyelination. Oligonucleotide agents, unlike therapeutic antibodies, can inhibit the expression of intracellular molecules, including non-coding RNAs involved in biological and pathological processes, with high specificity and selectivity. Emerging evidence on the genetics of many immunological diseases has enhanced not only our knowledge on complex disease pathogenesis but also on exploratory therapeutic targets. Additionally, we demonstrated that Toc-HDO can significantly silence the target endogenous gene in human cultured lymphocytes. Multiple nucleic acid therapeutics have been successfully developed over the past decade into approved treatment options. Clinical translation of HDO technology for human immunological disorders have come closer to reality, albeit further investigation is required. Developing HDO-mediated gene silencing technology based on manipulating lymphocyte functions will provide a new therapeutic platform to treat autoimmune diseases, inflammation, and cancer.

## Methods

**Design and synthesis of oligonucleotides**. A series of 16mer ASOs were designed to target mouse *Itga4* and *Dmpk* mRNA, *Malat1* RNA, and human *Stat3* mRNA. The ASOs had a 10 DNA nucleotide gapmer structure flanked by 3 LNA nucleotides. All internucleotide linkages were modified by phosphorothioate substitution to increase plasma ASO stability and protein binding, which ultimately increased tissue bioavailability[68]. *Itga4*-targeting ASO sequence: 5′-A^G^C^C^c^a^t^g^c^g^c^t^c^t^t^T^G^C^G-3′. *Itga4*-targeting scramble ASO sequence: 5′-G^C^C^G^A^a^c^g^c^g^t^t^g^a^c^T^C^T-3′ and 5′-G^C^G^C^A^G^g^c^g^a^t^c^t^t^a^t^g^C^T^C-3′. *Dmpk* mRNA, *Malat1* RNA, and *Stat3* mRNA targeting sequences: 5′-A^C^A^a^t^a^a^a^t^a^c^c^g^A^A^G^G-3′, 5′-C^T^A^g^t^t^c^a^c^t^g^a^a^T^G^C-3′, and 5'-C^T^A^t^t^t^g^g^a^t^g^t^c^A^G^C, respectively. *Malat1*-targeting scramble ASO sequence: 5′-A^C^G^t^g^a^t^c^g^c^c^t^t^A^T^A-3′. *Malat1*-targeting mismatch ASO sequence: 5′-C^T^G^g^t^g^c^a^g^t^g^a^a^T^G^C-3′. For all 16-mers, lowercase letters represent

DNA, bold uppercase letters represent LNA (capital C denotes LNA 5-methylcytosine), and carets represent phosphorothioate linkages. For flow cytometry and confocal microscopy, Alexa Fluor 647 or Cy5 were covalently bound to the ASO 5′-ends.

A series of 16-mer cRNAs were designed to be complementary to the ASO sequences. Phosphorothioate-modified 2′-O-methyl sugar modifications were used in cRNA complementary to the ASO strand LNA for protection from exonucleases. To produce Toc-cRNA, α-tocopherol was covalently bound to the 5′-end of the cRNA strand (Fig. 1). To generate Toc-HDO, equimolar amounts of ASO and cRNA strands in PBS were heated at 95 °C for 5 min and slowly cooled to room temperature. All ASOs and cRNAs were synthesized by Gene Design (Osaka, Japan) or Hokkaido System Science (Sapporo, Japan).

**Mouse studies**. Animal experiments were performed at Tokyo Medical and Dental University. All experimental protocols were approved by the Institutional Animal Care and Use Committee of Tokyo Medical and Dental University (No. 0170179A). Experimental procedures were in accordance with the ethical and safety guidelines for animal experiments of Tokyo Medical and Dental University. Male wild-type C57BL/6 mice aged 6–9 weeks and BALB/c mice aged 10 weeks (Oriental Yeast, Tokyo, Japan) were kept on a 12-h light/dark cycle in a pathogen-free animal facility with free access to food and water (temperature: 18–24 °C; humidity: 40–70%). Toc-HDO, ASO, or PBS were administered by tail-vein injection according to body weight. For postmortem analyses, mice were deeply anesthetized with intraperitoneally administered pentobarbital (60 mg/kg) and sacrificed by transcardiac perfusion with PBS after confirming an absent blink reflex.

**Induction of EAE**. To induce active EAE, 9–10-week-old female C57BL/6J mice were immunized subcutaneously with 200 μg MOG$_{35–55}$ peptide (MBL, Tokyo, Japan) and 400 μg *Mycobacterium tuberculosis* in incomplete Freund's adjuvant (Difco Laboratories). Pertussis toxin (200 ng per mouse; List Biologicals) was injected intraperitoneally the same day and 2 days post-immunization. Animals were observed daily for clinical symptoms and scored by a masked investigator as follows: 0, no clinical disease; 1, tail weakness; 2, tail paralysis; 3, hindlimb weakness; 4, forelimb weakness; 5, forelimb paralysis; 6, moribund or death. To induce adoptive-transferred EAE, donor mice were immunized as described. The mice were euthanized 12 days after immunization, and the draining lymph nodes and spleen were harvested. Single-cell suspensions were cultured in RPMI-1640 supplemented with 10% fetal bovine serum (FBS), 50 μM 2-mercaptoethanol, 2 mM L-glutamine, 100 U/ml penicillin, 100 μg/ml streptomycin, 10 μM MOG$_{35-55}$ peptide, and 3 μM Toc-HDO or ASO targeting *Itga4*. After 72 h, cells were harvested and resuspended in Hanks' balanced salt solution. A total of $1 \times 10^7$ viable cells in 100 μl were injected into the tail vein of wild-type mice. Mice also received two doses of pertussis toxin (200 ng per mouse) on the day of transfer and 2 days post-transfer.

**MHC-mismatched allogeneic hematopoietic cell transplantation and clinical assessment of acute GVHD**. MHC-mismatched allogeneic hematopoietic stem cell transplantation (B6 to BALB/c) was performed. Briefly, B6 mice (H-2b) were treated with PBS, *Itga4*-ASO, and Toc-HDO at corresponding doses of 50 mg/kg of ASO. Three days later, spleen T cells were isolated using a Pan T cell isolation Kit II for mouse (Miltenyi Biotec, Bergisch Gladbach, Germany). Bone marrow was flushed from non-treated B6 mice tibia and femur. TCD-BM were obtained using mouse CD3ε MicroBead Kit (Miltenyi Biotec) according to the manufacturer's instructions. In all, $5 \times 10^6$ TCD-BM along with $1 \times 10^6$ spleen T cells were transplanted into lethally irradiated (900 cGy on day −1) allogeneic BALB/c recipient mice (H-2d). Recipients were monitored daily for survival and clinical GVHD score as described previously[69]. The scoring system for acute GVHD had six clinical criteria (maximum index = 11). Weight loss of <10% was scored 0, of >10% and <25% was scored as 1, and of >25% was scored as 2. For gastrointestinal symptoms, the scoring system denoted 0 as normal and 1 as suffering from diarrhea. For posture and activity, the scoring system denoted 0 as normal, 1 for hunching at rest and a mild to moderate decrease in activity, and 2 for severe hunching and a severe decrease in activity. For fur texture and skin integrity, the scoring system denoted 0 as normal, 1 for mild to moderate fur ruffling and scaling of the paws and tails, and 2 for severe fur ruffling and an obviously denuded skin. GVHD experiments were performed with four mice per group

**In vivo studies**. Mouse peripheral blood, splenic, and thymic lymphocytes were isolated using Lymphocyte Separation Medium 1077 (PromoCell, Heidelberg, Germany). Briefly, peripheral blood and cell suspensions were obtained by crushing the spleen and thymus. Suspensions were diluted with an equal volume of sterile PBS and the diluted cell suspension was carefully overlaid on three volumes of Lymphocyte Separation Medium 1077 in 15 ml tubes and centrifuged at $400 \times g$ for 40 min without brakes. Mouse lymphocytes were removed from the liquid/medium interface and washed three times with 0.1% bovine serum albumin (BSA) in PBS.

**Cell culture**. The EL4 mouse T cell line and the Jurkat T cell line were obtained from the American Type Culture Collection and cultured as previously

described[70,71]. Briefly, cells were cultured in RPMI 1640 containing 10% FBS, 2 mM L-glutamine, 100 U/ml penicillin, 100 μg/ml streptomycin, and 50 μM 2-mercaptoethanol (all from Nacalai Tesque, Kyoto, Japan) in a humidified chamber at 37 °C containing 5% $CO_2$.

Mouse peripheral blood and lymph node T cells were expanded from the cell suspension described above. Cells were re-suspended in RPMI-1640 containing 10% FBS, 2 mM L-glutamine, 100 U/ml penicillin, 100 μg/ml streptomycin, and 50 μM 2-mercaptoethanol and incubated for 2 h in a humidified chamber at 37 °C containing 5% $CO_2$. Non-adherent cells were collected and incubated for another 2 h to remove monocytes. Finally, non-adherent cells were washed three times and cultured with 1 μg/ml anti-CD3 antibody (BD Pharmingen, clone 145-2C11, #553058) and 2 μg/ml anti-CD28 antibody (BD Pharmingen, clone 37.51, #557393) for 2–3 days.

**Treatment of cells with endocytosis inhibitors**. Inhibitors were used at the following concentrations: amiloride: 1–3 mM; chlorpromazine: 5–30 μM; dynasore: 10–100 μM; filipin III, 1–3 μM; and cytochalasin D, 10–20 nM (all from Sigma-Aldrich). Cells were pretreated with each inhibitor for 1 h at 37 °C and then treated with 500 nM ASO or Toc-HDO for 4 h at 37 °C in the presence of inhibitors. Cells were then washed and cultured in fresh media without inhibitors for 20 h prior to RNA isolation. In experiments where ASO or Toc-HDO treatment was performed at 4 °C, the cells were incubated at specified temperatures for 1 h and then treated with ASOs or Toc-HDO for 4 h at the same temperature. Cells were washed, replenished with fresh media, and incubated for another 20 h at 37 °C prior to RNA isolation.

**Stimulation of cytokine production**. Primary T cells isolated as above were seeded at $1 \times 10^6$ cells in a 24-well plate and incubated with 1 μM Toc-HDO or ASO targeting *Itga4* or *Malat1*, followed by supplementation with 50 ng/ml of phorbol myristate acetate and 500 ng/ml of ionomycin for 4 h at 37 °C with 5% $CO_2$. After incubation, the cell suspensions were centrifuged and supernatants were analyzed for the presence of interferon-γ and TNF-α using an enzyme-linked immunosorbent assay using paired antibodies (eBioscience).

**RNA isolation and quantitative real-time polymerase chain reaction (RT-PCR)**. Total RNA was extracted from single-cell suspensions following incubation with gene-specific ASO or Toc-HDO using ISOGEN (Nippon Gene, Tokyo, Japan). To detect mRNA, RNA (1 μg) was reverse transcribed with Transcriptor Universal cDNA Master Mix (Roche Diagnostics). To estimate mRNA expression, quantitative RT-PCR analysis was performed using the TaqMan MicroRNA Reverse Transcription Kit, a Light Cycler 480 Real-Time PCR Instrument, and Light cycler 480 software (Roche Diagnostics). The primers and probes for mouse *Dmpk* (NM_032418.2), *Epn2* (NM_001252188.2), *Gfap* (NM_001131020.1), *Glyceraldehyde-3-phosphate dehydrogenase* (*Gapdh*, NM_001289726.1), *Iba1* (also known as *Aif1*, NM_001361501.1), *Itga4* (NM_010576.4), *Interferon-β* (*Ifn-β*, NM_010510.1), *Interferon-γ* (*Ifn-γ*, NM_008337.4), *Il-1b* (NM_008361.4), *Malat1* (NR_002847.3), and *Tnf-α* (NM_013693.3) were purchased from Thermo Fisher Scientific. Relative target gene mRNA levels were normalized to *Gapdh* mRNA levels.

**Flow cytometry and fluorescence-activated cell sorting**. The delivery efficiency of ASO or Toc-HDO into primary T cells in vivo and EL4 in vitro cells was assayed by flow cytometry using Alexa Flour 647-labeled ASO. Cells were harvested at the indicated time points and washed in PBS by centrifugation at $500 \times g$ for 5 min. The cells were then washed and suspended in PBS with 2% BSA and 0.05% sodium azide (Sigma-Aldrich). Surface marker expression levels were determined using the following antibodies: CD3-PE (BioLegend, clone 17A2, 100205), CD45R/B220-PECy7 (BioLegend, clone RA3-6B2, #103221) or CD45R/B220-FITC (BioLegend, clone RA3-6B2, #103205), and CD49d-APC (BioLegend, clone R1-2, #103621). All related antibodies are listed in Supplementary Table 2. Cells were then sorted using a BD FACSAria cell sorter, BD FACSDiva software, and FlowJo v10 software (BD Biosciences) or analyzed using a BD FACSVerse cell analyzer and BD FACSuite software (BD Biosciences). The sorting/gating strategy is shown in Supplementary Fig. 2g.

**Western blot analysis**. Whole-cell extracts were isolated using RIPA lysis buffer (50 mM Tris-HCl pH 8.0, 150 mM NaCl, 1% Nonidet P-40, 0.5% sodium deoxycholate, 0.1% sodium dodecyl sulfate (SDS)) supplemented with protease inhibitors (Roche). The proteins were separated on a 15% SDS–polyacrylamide gel electrophoresis gel and electrophoretically transferred to polyvinylidene difluoride membranes (Millipore). Membranes were then incubated with the blocking buffer followed by incubation with the primary antibodies against Integrin α4 (Cell Signaling, clone D2E1, #8440), STAT3 (Cell Signaling, clone 124H6, #9139) or peroxidase-conjugated GAPDH (Wako, clone 5A12, #015-25473) and the appropriate secondary antibodies. Protein band intensities were quantified with the Fiji image processing software (National Institutes of Health)[72].

**Oligonucleotide concentration in lymphocytes**. Following isolation of peripheral blood lymphocytes from mice injected with Alexa Fluor 647-conjugated ASO or Toc-HDO, detection assays were performed on a microplate reader Infinite M1000 PRO (Tecan, Männedorf, Switzerland). Oligonucleotide concentration in a cell was determined by comparing the measured fluorescence intensity with that of the Alexa Fluor 647-labeled reference oligonucleotides of known concentration and corrected with division by total cell counts.

**Histopathological analyses of spinal cord tissues**. For pathological analyses, all tissues were fixed in 10% neutral-buffered formalin solution for 24 h, embedded in paraffin, and cut into 5-μm sections. IHC staining was performed with anti-CD4 polyclonal antibody (NOVUS Biologicals, #NBP1-19371) and detected using Histofine Simple Stain Mouse MAX PO (R) (Nichirei Biosciences, Tokyo, Japan). For immunofluorescence (IF), slides were stained with 4′,6-diamidino-2-phenylindole (DAPI; Vector Laboratories) to visualize the nuclei and were immunolabeled with antibodies against myelin basic protein (Abcam, # ab40390) and Iba1 (Wako, #019-19741). Samples were then incubated with Alexa Fluor 488-conjugated (Invitrogen, #A11006) or Alexa Fluor 647-conjugated (Invitrogen, #A21244) secondary antibody. All IHC or IF antibodies are listed in Supplementary Table 2. For studies using Cy5- or Alexa Fluor 647-conjugated ASO and Toc-HDO, mouse lymphocytes were centrifuged directly onto glass slides and stained with DAPI. All images were acquired with an A1R confocal laser scanning microscope (Nikon, Tokyo, Japan) and BZ-X700 fluorescence microscope (Keyence, Osaka, Japan). Quantitative image analysis was performed using the Fiji image processing software (National Institutes of Health).

**Statistical analysis**. All data represent mean ± s.e.m. Differences among three groups were analyzed by one-way analysis of variance followed by Holm's post hoc test. Statistical differences between two groups were analyzed using two-sided Student's two-tailed t test. Dose response data were fitted to a four-parameter log-logistic curve using GraphPad Prism 8.3.0 and Fiji image processing software.

**Reporting summary**. Further information on research design is available in the Nature Research Reporting Summary linked to this article.

## Data availability
The datasets from this study are provided in the Supplementary Information/Source data file. Source data are provided with this paper.

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

## Acknowledgements

We thank Dr. Kouhei Yamamoto and Professor Masanobu Kitagawa (Department of Comprehensive Pathology, Graduate School of Medicine and Dental Sciences, Tokyo Medical and Dental University, Tokyo, Japan) for assisting in microscope slide preparation. This research was supported by the Basic Science and Platform Technology Programs for Innovative Biological Medicine (18am0301003h0005) and Advanced Biological Medicine (20am0401006h0002) to T.Y. from the Japan Agency for Medical Research and Development (AMED; Tokyo, Japan) and a JSPS KAKENHI Grant-in-Aid for Scientific Research (S) (17H06109 to T.Y. and T.N.) and (A) (19H01016 to T.N. and T.Y.) from the Ministry of Education, Culture, Sports, Science, and Technology (MEXT) of Japan (Tokyo).

## Author contributions

M.O. designed the study, performed experiments, analyzed data, prepared figures, and wrote the manuscript; T.N. designed and performed experiments, interpreted data, and wrote the manuscript. K.I., R.N., K.Y.-T., H.M., A.A. and Y.M. performed experiments and analyzed data. C.A. and T.Y. supervised the research.

## Competing interests

T.Y. collaborates with Takeda Pharmaceutical Co., Ltd., Daiichi Sankyo Co., Ltd., and Rena Therapeutics Inc. and serves as an academic advisor for RENA Therapeutics Inc. and Braizon Therapeutics Inc. The other authors declare no competing interests.
