## [Peer Review File · Nature Communications]

Reviewers' Comments:

Reviewer #1:

Remarks to the Author:

This is a very interesting paper demonstrating the effects of intravenous injection of heteroduplex oligonucleotides (HDO) comprised of antisense oligonucleotide (ASO) and its complementary RNA conjugated to alpha-tocopherol (TOC) targeting *Itga4* (CD49d) in actively induced experimental autoimmune encephalomyelitis (EAE) in a prophylactic and therapeutic setting as well as in T cell transfer EAE. The effects of Toc-HDO are much stronger compared to ASO alone. The approach targets T cells. So far it is not clear if the effects are also seen in other models of autoimmunity, transplantation or infection.

Therefore, the paper would strongly gain by adding another autoimmune model (like arthritis or diabetes type 1), a transplantation model and an infection model. The authors should tackle the questions, if there is loss of protection from infection by this approach which could reduce its value for treatment of autoimmune disease and if there are effects of the approach independently of the used autoimmune model. Also, the authors should shortly address the question in the discussion about the possible additive value of this approach in translation for humans.

Reviewer #2:

Remarks to the Author:

In this study, Ohyagi and colleagues present a novel nucleotide delivery mechanism for the purpose of modifying lymphocytes *in vivo*. They show that this modification provides enhanced uptake and knockdown efficiency compared to current antisense oligonucleotide-based gene silencing strategies. They also demonstrate that their technique's enhanced gene knockdown capabilities can lead to improved disease outcomes in a mouse model of MS.

Overall, this is a valuable and promising study, especially given the current difficulties in modifying lymphocyte gene expression *in vivo*. However, there are some key controls and experimental details that are missing from the current study, making acceptance of this new technique difficult in its current form. These are detailed below:

Major comment: The authors pay considerable attention to resolving the mechanisms of how their HDO complexes are taken up by lymphocytes, and supported by the data in Figure 3. However, these data are currently not given sufficient emphasis in the ms and are a very important component of the study to support their proposed technique of 'passive' uptake of the complex by lymphocytes, e.g., Figs. 3D,E showing fluorophore-conjugated HDO uptake.

Major Issues:

- As currently presented, it is possible to interpret the entire study as HDO complexes non-specifically interfere with the lymphocytes they are taken up by. It is essential that the authors show that each of their HDO complexes are specific for the gene they target, i.e., that *Itga4*-HDO does not knock down *Malat1* and vice versa. Absent this information, it is possible that the HDO complex interferes with larger gene expression pathways and results in dysfunctional lymphocytes, which might also explain why HDO treatment leads to improved EAE outcomes. Similarly, it would be helpful to show that their *Itga4*-HDO-treated T cells are not globally dysfunctional. The authors should show that treated T cells have similar viability to controls, and should investigate whether they respond, as expected (e.g., TNF, IFN γ , etc) when treated with PMA-Ionomycin.
- In a similar vein, the three genes analyzed in this study (*Itga4*, *Malat1*, and *Dmpk*) are all very poorly expressed in resting T and B cells: *Itga4* is only expressed well on LCMV-activated CD8 T cells and mature NK cells; *Malat1* on CD8+ DCs; and *Dmpk* on red pulp macrophages and thymic epithelium. This is confirmed in Fig 2D, where *Itga4* staining is very low even on the PBS control mice. Evaluating the efficacy of HDO knockdown would be more convincing using a highly expressed gene for at least one experiment. For example, showing that an anti-CD44-HDO reduces CD44 expression on EL4 cells would potentially validate the small (albeit statistically measurable) changes in *Itga4* surface expression induced by HDO treatment.

Results/Methods:

1. To investigate the delivery mechanism of Toc-HDO into mouse lymphocytes, the authors treated mouse cultured lymphocytes with specific endocytosis pathway inhibitors and evaluated the expression of target genes by PCR, then further evaluated engulfment of fluorescence labeled-Toc-HDO by FACS. A direct analysis of lymphocyte uptake of fluorescence labeled-Toc-HDO in the presence of different endocytosis pathway inhibitors under fluorescence microscopy should be provided.

2. Additional data is needed to support the specificity of Itga4 mRNA-targeting Toc-HDO. Sequence BLAST analysis shows that ASO sequence can also target gene Epn2, which encodes a protein involved in clathrin-dependent endocytosis. Assessment of Epn2 mRNA levels following treatment of cells or mice with Itga4 Toc-HDO would improve this analysis.

3. The authors state that T cell and B cells were sorted using anti-CD3 antibody and anti-CD45RA antibody, respectively. However, CD45RA is mainly expressed by naive T cells. An appropriate marker for B cells should be B220/CD45R, as noted in the legend.

Minor Issues:

1. The authors nicely show that their HDO treatment is more effective than traditional antisense-oligonucleotide strategies. However, in both the introduction and discussion, the authors claim that their approach may be better than modern lipid nanoparticle, aptamer, and protein conjugate approaches, because these approaches have poor specificity for their target cell populations. The authors also claim that their approach is lymphocyte-specific, but only test it in lymphocytes, and do not include the proper controls to support this claim. The ms should be revised to either remove claims of lymphocyte specificity or include analyses of non-lymphoid tissues in mice with intravenous-HDO treatments that are not impacted by the HDO treatment.

2. The authors should review citations. For example, (p 5, line 63-65: "Other gene silencing techniques commonly ... interfering RNA delivery systems (8, 9, 10). However, these methods also yield low transfection efficacy, especially in vivo". These references contain no aptamer-related studies.

The authors state that other methods yield low transfection efficacy. In fact, gene silencing efficiency can be high (~80%), even in CD4-specific T cells, using an aptamer-based approach (Wheeler et al. JCI, 2011).

3. HDOs have been conjugated with α -tocopherol to facilitate delivery to lymphocytes via binding to serum lipoproteins in blood and distribution via the α -tocopherol transport pathway. Since α -tocopherol preferentially binds to serum lipoproteins, will this approach result in relatively low HDO delivery efficiency to spleen and other lymphoid organs? Moreover, α -tocopherol is physiologically essential to enhance proliferation and form effective immune synapses by lymphocytes, opening the possibility of potential side effects to HDO- α -tocopherol.

Reviewer #3:

Remarks to the Author:

In this manuscript, the authors developed heteroduplex oligonucleotides (HDOs) which consists of a gapmer strand and a complementary RNA strand with α -tocopherol (Toc) conjugation as a delivery ligand for lymphocytes. Then they tested and compared the potency, adverse effects, and gene silencing duration of Toc-HDOs and the gapmer strand ASOs targeting ITGA4, MALAT1 and DMPK in different tissues of lymphocytes in vivo. In addition, they used in vitro model and specific endocytosis pathway inhibitors to elucidate the mechanism of endocytosis by Toc-HDOs. Finally, they applied ITGA4-targeting Toc-HDO in both adoptive transfer and active experimental autoimmune encephalomyelitis models and showed target gene reduction and mitigated symptoms. Overall, this work is of interest to the oligonucleotide therapeutics field and is well-design. Therefore, I recommend this article for publication after the concerns below are properly addressed.

1. The authors compared Toc-HDOs and the parent gapmer ASOs. Have the authors looked at gapmer ASOs conjugated with α -tocopherol (Toc)?

2. For the screening of sequences in gapmers targeting ITGA4, MALAT1 and DMPK in vitro, have the authors used negative control oligonucleotides (mismatch and scrambled controls) to validate the on-target effects? For example, in Figure S1A, using WT as negative control is not sufficient.

3. In line 195 and 298, the authors mentioned that besides macropinocytosis and clathrin-independent dynamin-dependent pathways, Toc-HDO is also uniquely uptaken by a clathrin- and dynamin-independent pathway.

1) This statement is based on Fig 4D. However, the authors have not tested concentration larger than 100 μ M dynasore. Will higher concentration reverse the inhibition of MALAT1 by Toc-HDO?

2) The authors should list the IC50 of the five specific endocytosis pathway inhibitors to rationalize the different concentration range used in Fig 4 and Fig S2.

4. In Fig 2D and Fig S1C, the authors used FACS to quantify the inhibition of ITGA4 protein in sorted T cells and B cells. Is western blot feasible in this protocol? Having gel image as primary data could validate the gene silencing in both mRNA and protein levels.

5. In Fig 2A-C and Fig S1B, the selection of tissues is different. The authors need to provide rationale for it.

1) In Fig 2A-B, the authors selected lymph nodes for ITGA4, and thymic lymphocytes for MALAT1.

2) In Fig S1B, the authors selected three tissues instead of four which shown in Fig 2A-B.

3) In Fig 2A and 2C, is peripheral blood equal to peripheral lymphocytes? It's a good idea to mention what types of tissues consist of lymphocytes in vivo for general audience.

6. In line 731, Fig 4 legend, the concentration should be 500 nM instead of 500 μ M.

Response to the Reviewer' comments

Reviewer #1:

We wish to express our appreciation to the reviewer for his/her insightful comments, which have helped us significantly improve the paper.

Comment 1: *it is not clear if the effects are also seen in other models of autoimmunity, transplantation or infection. Therefore, the paper would strongly gain by adding another autoimmune model (like arthritis or diabetes type 1), a transplantation model and an infection model.*

Response: We would like to thank the review for this insight. To investigate additional therapeutic application of HDO technology, we examined therapeutic efficacy of Toc-HDO targeting *Itga4* in a fully MHC-mismatched acute graft versus host disease (GVHD) mouse model. The previous studies reported the role of *Itga4* in the pathology of GVHD (*Blood* 2005;105:4191–4199.) and *Itga4* deletion results in significant improvement in GVHD and survival (*Leukemia* 2020;34:3100–3104.). We first isolated T cell-depleted bone marrow cells (TCD-BM) from non-treated B6 mouse. TCD-BM were intravenously injected with spleen T cells isolated from mice 72 h after intravenous injection of *Itga4*-targeting Toc-HDO, ASO, or PBS into lethally irradiated allogeneic BALB/c recipient mice. Peak GVHD clinical score in mice injected with Toc-HDO-treated T cells was significantly lower than mice injected with PBS- and ASO-treated T cells. Survival rate tended to be higher in Toc-HDO-treated T cells group compared with PBS- and ASO-treated T cells group. These findings support clinical application of Toc-HDO in transplantation. We have deleted “Finally,” (page 12, line 199), and added the following paragraph (page 14, line 233) and cited those papers as reference number 38–40:

“Finally, we also examined HDO-mediated *Itga4* regulation in acute graft versus host disease (GVHD) model. GVHD remains a major cause of morbidity and mortality in allogeneic hematopoietic cell transplantation. A fully MHC-mismatched transplantation with T cell-depleted bone marrow cells (TCD-BM) along with spleen T cells results in GVHD, but not transplantation with only TCD-BM. In this model, the crucial role of *Itga4* in the pathology of GVHD^{38,39} offers inhibition of *Itga4* in allo-reactive T cells migration to peripheral tissues as a therapeutic approach for GVHD. We initially isolated spleen T cells from B6 mice intravenously injected with PBS, ASO, or Toc-HDO targeting *Itga4*. Then, we intravenously injected these spleen T cells along with TCD-BM isolated from non-treated B6 mice into lethally irradiated allogeneic BALB/c recipient mice. Mice transplanted with only TCD-BM did not develop GVHD as previously reported^{39,40} (Fig.

S6C). In contrast, daily monitoring revealed that significant lower GVHD clinical score in mice injected with Toc-HDO-treated T cells than PBS- and ASO-treated T cells (Fig. S6D). Survival rate tended to be higher in Toc-HDO-treated T cells group compared with PBS- and ASO-treated T cells group, but not significant (Fig. S6C). Taken together, modulation of lymphocyte function by Toc-HDO technology is a potent therapeutic strategy for autoimmune diseases, inflammation, and transplantation.”

Fig. S6C and D

(C) Survival curve and (D) maximum clinical GVHD score of BALB/c recipients after transplant with T cell-depleted C57BL/6J bone marrow (TCD) and splenic T cells treated with *Itga4*-targeting Toc-HDO, ASO, or PBS (N = 4 per group). Data are expressed as mean values \pm s.e.m. *p* values were calculated using one-way ANOVA with Holm’s post-test or log-rank test for survival curves (**p* < 0.05, ***p* < 0.01, ****p* < 0.001).

We have added the following phrase (page 12, line 197):

“Toc-HDO-mediated suppression of *Itga4* prevents experimental autoimmune encephalomyelitis and graft versus host disease.”

Furthermore, we have added the following text in materials and methods section (page 22, line 368) and reference 69:

“**MHC mismatched allogeneic hematopoietic cell transplantation and clinical assessment of acute graft versus host disease (GVHD)**

MHC mismatched allogeneic hematopoietic stem cell transplantation (B6 to BALB/c) was performed. Briefly, B6 mice (H-2b) were treated with PBS, *Itga4*-ASO, and Toc-HDO at corresponding doses of 50 mg/kg of ASO. Three days later, spleen T cells were isolated using a Pan T cell isolation Kit II for mouse (Miltenyi Biotec, Bergisch Gladbach, Germany). Bone marrow was flushed from non-treated B6 mice tibia and femur. T cell-depleted bone marrow cells (TCD-BM) were obtained using mouse CD3ε MicroBead Kit (Miltenyi Biotec) according to the manufacturer's instructions. 5×10^6 TCD-BM along with 1×10^6 spleen T cells were transplanted into lethally irradiated (900 cGy on day -1) allogeneic BALB/c recipient mice (H-2d). Recipients were monitored daily for survival and clinical GVHD score as described previously⁶⁹. GVHD experiments were performed with four mice per group."

Comment 2: *The authors should tackle the questions, if there is loss of protection from infection by this approach which could reduce its value for treatment of autoimmune disease and if there are effects of the approach independently of the used autoimmune model.*

Response: The reviewer's comment is reasonable. Depending on a target gene, HDO-based gene knockdown could induce immunosuppression as on-target effect, just like progressive multifocal leukoencephalopathy whose risk increases in patients treated with *Itga4*-blockade. On the other hand, if HDO complex compromises lymphocyte functions throughout, gene expressions would decrease and HDO treatment would ameliorate EAE symptoms with loss of protection from infection. We then evaluated the possibility that HDO treatment non-specifically impairs lymphocyte viability and functions. We first examined cell viability after HDO treatment in vitro. Primary T cells were treated with 1 μ M *Itga4*-targeting Toc-HDO or ASO for 24 h before staining with fixable viability dyes. Flow cytometry analysis showed no differences in cell viability of between HDO- and ASO-treated T cells. *Malat1*-targeting oligonucleotides showed the same results (Fig. S6A).

Fig. S6A

(A) Percent of cell viability measured by flow cytometry analysis using fixable viability dyes staining after 24 h incubation with 1 μ M of *Itga4*- or *Malat1*-targeting Toc-HDO, ASO, or PBS alone.

To assess if Toc-HDO suppresses lymphocyte responses to infection or inflammation, we evaluated IFN γ and TNF α production of ASO or Toc-HDO-treated primary T cells after stimulation with PMA and ionomycin. Primary T cells were treated with *Itga4*- or *Malat1*-targeting Toc-HDO or ASO for 24 h, followed by stimulation with 50 ng/mL PMA and 500 ng/mL Ionomycin for 4 h. After centrifugation, IFN γ and TNF α levels in the supernatant were measured by ELISA. There is no decrease in IFN γ and TNF α levels in T cells treated with *Itga4*- and *Malat1*-Toc-HDO and ASO (Fig. S6B).

Fig. S6B

(B) ELISA analysis of IFN γ and TNF α production by primary T cells treated with 1 μ M of *Itga4*- or *Malat1*-targeting Toc-HDO or ASO for 24 h, followed by 4 h incubation with PMA and ionomycin.

These findings suggest that Toc-HDO treatment selectively suppresses target genes and does not inhibit lymphocyte functions, including the response to infection. We have added IFN γ and TNF α levels in T cells treated with Toc-HDO and ASO targeting *Itga4* and *Malat1* followed by PMA and ionomycin in Fig S6B. We have added the following text (page 13, line 205):

“We beforehand confirmed that *Itga4*-targeting ASO and Toc-HDO have no effect on cell viability and dysfunctional cytokine production (Fig. S6A and B).”

We have added the following text (page 24, line 401):

“Stimulation of cytokine production

Primary T cells isolated as above were seeded at 1×10^6 cells in a 24-well plate and incubated with 1 μ M Toc-HDO or ASO targeting *Itga4* or *Malat1*, followed by supplementation with 50 ng/mL of PMA and 500 ng/mL of ionomycin for 4 h at 37°C with 5% CO₂. After incubation, the cell suspensions were centrifuged, and supernatants were analyzed for the presence of IFN γ and TNF α using an enzyme-linked immunosorbent assay (ELISA) using paired antibodies (eBiosciences).”

Comment 3: *Also, the authors should shortly address the question in the discussion about the possible additive value of this approach in translation for humans.*

Response: The reviewer’s comment is reasonable, and we thank you for giving an opportunity to describe on the translational potential of HDO technology for human immunological diseases. In further considering the application to humans, we have examined the gene knockdown effect on human *Stat3*, a target gene for inflammatory diseases or lymphoma. Our additional study showed Toc-HDO can significantly reduce *hStat3* expression in human cultured lymphocytes at both mRNA and protein levels compared with ASO (Fig S5A and B).

Fig S5A and B

(A) Target *Stat3* mRNA levels measured by quantitative RT-PCR analyses in Jurkat cells 24 h after treatment with 100 nM or 1 μ M of *Stat3*-targeting Toc-HDO, ASO, or PBS alone without any transfection reagents. (B) STAT3 protein expression determined by western blot in Jurkat cells after treatment with *Stat3*-targeting Toc-HDO or ASO for 24 h. Quantitative RT-PCR data and band intensity shown are relative to *Gapdh* mRNA and GAPDH protein levels, respectively. Data are expressed as mean values \pm s.e.m. and are represented two independent experiments. *p* values were calculated using one-way ANOVA with Holm’s post-test ($*p < 0.05$, $**p < 0.01$, $***p < 0.001$).

We have added the following text (page 12, line 196) and cited the paper (Reilley MJ, *et al. J Immunother Cancer* **6**, 119 (2018)) as reference 36:

“For investigation into gene silencing effect of Toc-HDO in human lymphocytes, we evaluated target gene knockdown by Toc-HDO in human cultured lymphocyte. We targeted *Stat3*, which acts as a transcription activator downstream of many cytokines and growth factors receptors expressed on lymphocytes, because specific ASO for *Stat3* has high efficacy and safety in previous clinical studies³⁶. Toc-HDO targeting *Stat3* demonstrated significant target gene knockdown at a high dose in Jurkat cells compared to ASO (Fig. S5A). At protein levels, Toc-HDO reduced STAT3 expression (Fig S5B). These findings support clinical application of Toc-HDO in regulating human lymphocyte functions.”

We have also added the following text in the discussion section to describe on the translational potential of HDO technology for human immunological diseases. (page 19, line 312):

“Additionally, we demonstrated that Toc-HDO can significantly silence the target endogenous gene in human cultured lymphocytes. Multiple nucleic acid therapeutics have been successfully developed over the past decade into approved treatment options. Clinical translation of HDO technology for human immunological disorders to come closer to reality, albeit further investigation is required. Developing HDO-mediated gene silencing technology based on manipulating lymphocyte functions will provide a new therapeutic platform to treat autoimmune diseases, inflammation, and cancer.”

Reviewer #2:

We wish to express our appreciation to the reviewer for his/her insightful comments, which have helped us significantly improve the paper.

Comment 1: *The authors pay considerable attention to resolving the mechanisms of how their HDO complexes are taken up by lymphocytes, and supported by the data in Figure 3. However, these data are currently not given sufficient emphasis in the ms and are a very important component of the study to support their proposed technique of 'passive' uptake of the complex by lymphocytes, e.g., Figs. 3D,E showing fluorophore-conjugated HDO uptake.*

Response: We would like to thank the review for this insight. As the reviewer pointed out, our findings shown in Fig. 3D and E indicated that fluorophore-conjugated HDO was taken up by lymphocytes through “passive targeting” mechanisms that α -tocopherol conjugation enhances HDO’s binding to plasma proteins to prevent hepatic and renal clearance and increase blood retention, rather than directly binding to cellular surface receptor, or “active targeting”. Then, we have added the blood retention time of fluorophore-conjugated Toc-HDO and ASO (Fig. 3C) showing that Toc-HDO exhibited longer blood retention than ASO even 12 h after injection. HDO conjugated with α -tocopherol allows longer retention time in the blood through binding to plasma protein as we previously reported (12) and, at cellular levels, more efficient cellular uptake by several uptake mechanisms as shown in the present manuscript. We have added the following phrase (page 9, line 140):

Fig. 3C

(C) Time course of oligonucleotide concentration in plasma and peripheral blood lymphocytes after injection of 20 mg/kg Alexa Fluor 647 (AF647)-labeled Toc-HDO or ASO. Oligonucleotide concentration was calculated by fluorescence intensity.

“We injected 20 mg/kg ASO oligonucleotides with 5’ Alexa Fluor 647 labels into the tail vein and examined their presence in plasma and lymphocytes 1, 3 6, 12, and 24 h after injection. Significant retention of Alexa Fluor 647-labeled oligonucleotides was observed in plasma at 12 h and lymphocytes treated with Toc-HDO even at 24 h after injection (Fig. 3C).”

We have further added the following text (page 10, line 156):

“Collectively, intravenously-administered Toc-HDO induces continuous reduction of target gene expression through increased blood retention, is more quickly taken up and localized to peripheral blood lymphocyte nuclei, and persists longer than ASO at cellular levels.”

We have added the following text (page 12, line 196):

“These mechanisms were supported by the findings of cellular uptake analyses using fluorescence-conjugated oligonucleotides (Fig. 3D and E). Flow cytometry and fluorescent microscopy revealed that although lymphocytes take up ASO, their cellular uptake of Toc-HDO is more quickly and efficiently, indicating that α -tocopherol conjugation induces a passive targeting effect that facilitates ligand binding to serum proteins in order to decrease clearance and increase circulation time (Fig. 3C), rather than an active targeting effect that a ligand binds to a specific receptor expressed on the target cells. HDO conjugated with α -tocopherol allows prolonged blood retention time and efficient internalization through several uptake pathways at cellular levels.”

And we have added the following text (page 16, line 272):

“Conjugation with α -tocopherol allows HDO to bind to serum lipoprotein such as HDL and LDL, resulting in an efficient delivery along with physiological pathway of α -tocopherol¹². Unconjugated HDO as well as ASO is small enough to be rapidly filtered out of blood by the kidney and excreted in urine, so binding to serum proteins enhanced by ligand conjugation maintains HDO in circulation long enough to distribute broadly to peripheral tissues. Efficient cellular uptake of Toc-HDO was attributed to multiple roles of α -tocopherol conjugation, including highly retention in blood and escape from renal excretion with the passive targeting effect, and efficient internalization through multiple cellular uptake mechanisms.”

Comment 2: As currently presented, it is possible to interpret the entire study as HDO complexes non-specifically interfere with the lymphocytes they are taken up by. It is essential that the authors show that each of their HDO complexes are specific for the gene they target, i.e., that *Itga4*-HDO does not knock down *Malat1* and vice versa. Absent this information, it is possible that the HDO complex interferes with larger gene expression pathways and results in dysfunctional lymphocytes, which might also explain why HDO treatment leads to improved EAE outcomes. Similarly, it would be helpful to show that their *Itga4*-HDO-treated T cells are not globally dysfunctional. The authors should show that treated T cells have similar viability to controls, and should investigate whether they respond, as expected (e.g., TNF, IFN γ , etc) when treated with PMA-Ionomycin.

Response: The reviewer’s comment is reasonable. In accordance with the reviewer’s comment, we examined *Malat1* expression in lymph node lymphocytes 72 h after intravenous injection of *Itga4*-Toc-HDO or ASO. *Itga4*-HDO treatment does not inhibit *Malat1* RNA expression. *Malat1*-HDO does not reduce *Itga4* mRNA expression. We have added these results as Fig. S2B and the following text (page 7, line 105):

“We further checked whether off-target effects occur between each of target sequences, and expression of *Epn2* mRNA which is 14 bp matched with *Itga4*-targeted ASO sequence; however, each of Toc-HDO and ASO targeting *Itga4* and *Malat1* was specific for the target gene (Fig.S1B and Fig.S2B).”

Fig. S2B

(B) Quantitative RT-PCR analyses of *Malat1* RNA levels in mouse lymph node lymphocytes 72 h after intravenous injection of 50 mg/kg *Itga4*-targeting Toc-HDO, ASO, or PBS alone, and *Itga4* mRNA levels in mouse lymph node lymphocytes 72 h after intravenous injection of 50 mg/kg *Malat1*-targeting Toc-HDO, ASO, or PBS alone. NS, not significant.

We next examined cell viability after HDO treatment in vitro. Primary T cells were treated with 1 μ M *Itga4*-Toc-HDO or ASO for 24 h before staining with fixable viability dyes. Flow cytometry analysis showed no differences in cell viability of between HDO- and ASO-treated T cells (Fig. S6A).

Next, in accordance with the reviewer's comment, we evaluated IFN γ and TNF α production of ASO or Toc-HDO-treated primary T cells after stimulation with PMA and ionomycin. Primary T cells were treated with *Itga4*- or *Malat1*-targeting Toc-HDO or ASO for 24 h, followed by stimulation with 50 ng/mL PMA and 500 ng/mL Ionomycin for 4 h. After centrifugation, IFN γ and TNF α levels in the supernatant were measured by ELISA. There is no decrease in IFN γ and TNF α levels secreted by T cells treated with both *Itga4*- and *Malat1*-Toc-HDO (Fig. S6B).

Fig. S6

(A) Percent of cell viability measured by flow cytometry analysis using fixable viability dyes staining after 24 h incubation with 1 μ M of *Itga4*- or *Malat1*-targeting Toc-HDO, ASO, or PBS alone. **(B)** ELISA analysis of IFN γ and TNF α production by primary T cells treated with 1 μ M of *Itga4*- or *Malat1*-targeting Toc-HDO or ASO for 24 h, followed by 4 h incubation with PMA and ionomycin.

As we showed in the previous manuscript, neither *Itga4*- nor *Malat1*-Toc-HDO induced high cytokine expressions in lymphocytes in vivo compared with ASO (Fig S2D and E). Based on our additional results, Toc-HDO treatment does not make T cells dysfunctional. We have added IFN γ and TNF α levels in T cells treated with Toc-HDO and ASO targeting *Itga4* and *Malat1* followed by PMA and ionomycin stimulation in Fig S6B. We have added the following text (page 13, line 205):

“We beforehand confirmed that *Itga4*-targeting ASO and Toc-HDO have no effect on cell viability and dysfunctional cytokine production (Fig. S6A and B).”

We have added the following text (page 24, line 401):

“Stimulation of cytokine production

Primary T cells isolated as above were seeded at 1×10^6 cells in a 24-well plate and incubated with 1 μ M Toc-HDO or ASO targeting *Itga4* or *Malat1*, followed by supplementation with 50 ng/mL of PMA and 500 ng/mL of ionomycin for 4 h at 37 $^{\circ}$ C with 5% CO $_2$. After incubation, the cell suspensions were centrifuged and supernatants were analyzed for the presence of IFN γ and TNF α using an enzyme-linked immunosorbent assay (ELISA) using paired antibodies (eBiosciences).”

Comment 3: *In a similar vein, the three genes analyzed in this study (Itga4, Malat1, and Dmpk) are all very poorly expressed in resting T and B cells: Itga4 is only expressed well on LCMV-activated CD8 T cells and mature NK cells; Malat1 on CD8+ DCs; and Dmpk on red pulp macrophages and thymic epithelium. This is confirmed in Fig 2D, where Itga4 staining is very low even on the PBS control mice. Evaluating the efficacy of HDO knockdown would be more convincing using a highly expressed gene for at least one experiment. For example, showing that an anti-CD44-HDO reduces CD44 expression on EL4 cells would potentially validate the small (albeit statistically measurable) changes in Itga4 surface expression induced by HDO treatment.*

Response: As the reviewer pointed out, *Dmpk* mRNA expression was low in mouse lymphocytes, especially peripheral blood lymphocytes. However, *Malat1* expresses ubiquitously in all types of lymphocytes at high values (<https://www.immgen.org/>, *Nat. Immunol.* 9, 1091–1094 (2008)). We examined *Cd44* mRNA expression in mouse lymphocytes, but *Cd44*, which is usually used as an activation marker, poorly expresses in resting lymphocytes as shown in Response Fig. 1.

Response Fig. 1

Gene expression levels measured by quantitative RT-PCR in mouse peripheral blood lymphocytes. Data are normalized to *Cd44* mRNA levels and expressed as values \pm s.e.m. mRNA expression level of *Cd44* was poorly expressed in resting lymphocytes.

Thus, we evaluated gene silencing efficacy of Toc-HDO targeting human *Stat3* which acts as a transcription activator downstream of many cytokines receptors expressed on lymphocytes. The clinical study of *Stat3*-ASO in patients with relapsed or treatment refractory non-Hodgkin's lymphoma showed well tolerated and subtle efficacy (NCT01563302, *J Immunother Cancer*. 2018;6(1):119.). We found the enough expression level of *Stat3* in Jurkat cells and significant *Stat3* mRNA reduction by treatment with Toc-HDO compared to ASO. Furthermore, western blots demonstrated significant STAT3 protein reduction by Toc-HDO. We have added the figures showing *Stat3* mRNA expression (Fig. S5A) and western blots showing STAT3 protein expression (Fig. S5B) in Jurkat cells treated with Toc-HDO, ASO, or PBS alone. We have added the following text (page 12, line 196):

“we evaluated target gene knockdown by Toc-HDO in human cultured lymphocyte. We targeted *Stat3*, which acts as a transcription activator downstream of many cytokines and growth factors receptors expressed on lymphocytes, because specific ASO for *Stat3* has high efficacy and safety in previous clinical studies ³⁶. Toc-HDO targeting *Stat3* demonstrated significant target gene knockdown at a high dose in Jurkat cells compared to ASO (Fig. S5A). At protein levels, Toc-HDO reduced STAT3 expression (Fig S5B).”

Fig S5

(A) Target *Stat3* mRNA levels measured by quantitative RT-PCR analyses in Jurkat cells 24 h after treatment with 100 nM or 1 μM of *Stat3*-targeting Toc-HDO, ASO, or PBS alone without any transfection reagents. (B) STAT3 protein expression determined by western blot in Jurkat cells after treatment with *Stat3*-targeting Toc-HDO or ASO for 24 h. Quantitative RT-PCR data and band intensity shown are relative to *Gapdh* mRNA and GAPDH protein levels, respectively. Data are expressed as mean values ± s.e.m. and are represented two independent experiments (N = 3). *p* values were calculated using one-way ANOVA with Holm’s post-test (**p* < 0.05, ***p* < 0.01, ****p* < 0.001).

We have added the following text (page 20, line 319):

“A series of 16mer ASOs were designed to target mouse *Itga4* and *Dmpk* mRNA, *Malat1* RNA, and human *Stat3* mRNA. The ASOs had a 10 DNA nucleotide gapmer structure flanked by 3 LNA nucleotides. All internucleotide linkages were modified by phosphorothioate substitution to increase plasma ASO stability and protein binding, which ultimately increased tissue bioavailability⁵⁹. *Itga4*-targeting ASO sequence: 5'-A^G^C^c^a^t^g^c^g^c^t^c^t^T^G^G-3'. *Dmpk* mRNA, *Malat1* RNA, and *Stat3* mRNA targeting sequences: 5'-A^C^A^a^t^a^a^t^a^c^g^A^G^G-3', 5'-C^T^A^g^t^t^c^a^c^t^g^a^a^T^G^C-3', and 5'-C^T^A^t^t^t^g^g^a^t^g^t^c^A^G^C, respectively.”

We have added the following phrase in materials and methods section (page 23, line 378).

“The EL4 mouse T-cell line and the Jurkat T-cell line were obtained from the American Type Culture Collection and”

Comment 4: *To investigate the delivery mechanism of Toc-HDO into mouse lymphocytes, the authors treated mouse cultured lymphocytes with specific endocytosis pathway inhibitors and evaluated the expression of target genes by PCR, then further evaluated engulfment of fluorescence labeled-Toc-HDO by FACS. A direct analysis of lymphocyte uptake of fluorescence labeled-Toc-HDO in the presence of different endocytosis pathway inhibitors under fluorescence microscopy should be provided.*

Response: In according to the reviewer's comment, we examined lymphocyte cellular uptake of fluorescence-labeled Toc-HDO and ASO by using fluorescence microscopy. Cultured EL4 cells were treated with each inhibitor for 1 h, followed by incubation with AF647-labeled Toc-HDO or ASO. After 30 min incubation, we observed cellular uptake of AF647-labeled oligonucleotides under fluorescence microscopy (BZ-X700, Keyence). Quantitative analysis showed cellular uptake inhibition by amiloride, micropinocytosis inhibitor, and dynasore, dynamin inhibitor, but no effects in the presence of other inhibitors. These results were consistent with our findings using flow cytometry. We have added representative images and quantitative analysis in Fig S3E and the following phrase (page 12, line 190):

“To further validate the mechanisms that internalize Toc-HDO molecules in EL4 cells, we measured Alexa Fluor 647-conjugated oligonucleotides in the presence or absence of amiloride or dynasore by flow cytometry and fluorescence microscope. Amiloride and dynasore both abolished ASO and Toc-HDO internalization (Fig. S3C, D and E)”

Fig S3E

(E) Representative images and quantitative data of AF647-labeled Toc-HDO or ASO internalized by EL4 cells treated with each inhibitor. Sections were stained with DAPI. Red, AF647; blue, DAPI. Scale bars, 50 μm .

We have added the following text in materials and methods section (page 27, line 449):

“All images were acquired with an A1R confocal laser scanning microscope (Nikon, Tokyo, Japan) and BZ-X700 fluorescence microscope (Keyence, Osaka, Japan). Quantitative image analysis was performed using Fiji image processing software (National Institutes of Health).”

Comment 5: Additional data is needed to support the specificity of *Itga4* mRNA-targeting Toc-HDO. Sequence BLAST analysis shows that ASO sequence can also target gene *Epn2*, which encodes a protein involved in clathrin-dependent endocytosis. Assessment of *Epn2* mRNA levels following treatment of cells or mice with *Itga4* Toc-HDO would improve this analysis.

Response: We would like to thank the review for this insight. We examined *Epn2* mRNA expression in mouse primary T cells treated *Itga4*-tageting Toc-HDO or ASO. Primary T cells were treated with 1 μM of *Itga4*-targeting Toc-HDO or ASO for 24 h. Neither Toc-HDO nor ASO induced significant *Epn2* mRNA reduction as an off-target effect. We have added *Epn2* mRNA levels in Fig. S1D. Furthermore, we have added the following text (page 7, line 105):

“We further checked whether off-target effects occur between each of target sequences, and expression of *Epn2* mRNA which is 14 bp matched with *Itga4*-targeted ASO sequence; however, each of Toc-HDO and ASO targeting *Itga4* and *Malat1* was specific for the target gene (Fig.S1D and S2B).”

Fig. S1D

(D) *Epn2* mRNA levels measured by quantitative RT-PCR in primary T cells 24 h after treatment with 1 μ M *Itga4*-targeting Toc-HDO, ASO, and PBS alone. NS, not significant.

We have added the following phrase in materials and methods section (page 24, line 409):

“the primers and probes for mouse *dystrophia myotonica-protein kinase* (*Dmpk*, NM_032418.2), Epsin 2 (*Epn2*, NM_001252188.2), Glial fibrillary acidic protein (*Gfap*, NM_001131020.1), *Glyceraldehyde-3-phosphate dehydrogenase* (*Gapdh*, NM_001289726.1),”

Comment 6: *The authors state that T cell and B cells were sorted using anti-CD3 antibody and anti-CD45RA antibody, respectively. However, CD45RA is mainly expressed by naïve T cells. An appropriate marker for B cells should be B220/CD45R, as noted in the legend.*

Response: We thank the reviewer for this pertinent comment. We used anti-CD45R/B220 antibody for B cell isolation in this study. We have changed from “anti-CD45RA” to “anti-CD45R/B220” (page 8, line 109), and “CD45RA” to “B220” in Fig 3D.

Fig 3D

(D) Representative histogram and quantitative data of AF647-labeled Toc-HDO or ASO internalized by T cell and B cell from peripheral blood 6 h after intravenous injection. MFI, mean fluorescence intensity.

Comment 7: *The authors nicely show that their HDO treatment is more effective than traditional antisense-oligonucleotide strategies. However, in both the introduction and discussion, the authors claim that their approach may be better than modern lipid nanoparticle, aptamer, and protein conjugate approaches, because these approaches have poor specificity for their target cell populations. The authors also claim that their approach is lymphocyte-specific, but only test it in lymphocytes, and do not include the proper controls to support this claim. The ms should be revised to either remove claims of lymphocyte specificity or include analyses of non-lymphoid tissues in mice with intravenous-HDO treatments that are not impacted by the HDO treatment.*

Response: The reviewer’s comment is reasonable. The previous manuscript contained misleading expressions that Toc-HDO could induce lymphocyte-specific gene knockdown. As described in the introduction, we previously reported that Toc-HDO achieves highly efficient gene silencing in liver and brain microvascular endothelial cells in vivo^{12,13}. Our present study demonstrated that Toc-HDO can highly reduce the target gene expressions in lymphocytes which have been considered to be hard-to-transfect without any transfection reagents, though not specific for lymphocytes. We have added the following text in discussion section (page 16, line 272):

“HDO-based technology successfully results in target gene knockdown in liver and brain endothelial cells^{12, 13}, not necessarily specific to lymphocytes.”

We further changed the following phrase in abstract (page 3, line 35):

“Our findings reveal the advantages of HDO with enhanced gene knockdown effect and different delivery mechanisms compared with ASO. Thus, regulation of lymphocyte functions by HDO is a potential therapeutic option for immune-mediated diseases.”

Comment 8: *The authors should review citations. For example, (p 5, line 63-65: “Other gene silencing techniques commonly ... interfering RNA delivery systems (8, 9, 10). However, these methods also yield low transfection efficacy, especially in vivo”. These references contain no aptamer-related studies. The authors state that other methods yield low transfection efficacy. In fact, gene silencing efficiency can be high (~80%), even in CD4-specific T cells, using an aptamer-based approach (Wheeler et al. JCI, 2011).*

Response: We would like to thank the reviewer for this insight. In accordance with the reviewer’s comment, we have cited CD4-aptamer-based approach by Wheeler et al. (reference 11) in page 5, line 65 and deleted the following text (page 5, line 65):

“However, these methods also yield low transfection efficacy, especially in vivo.”

Alternatively, we have added the following text (page 5, line 65):

“These studies support the development of targeted oligonucleotide delivery platforms for therapeutics of lymphocyte-based diseases.”

Furthermore, we have deleted the following text (page 15, 251):

“However, these technologies also have limitations, such as immune-related adverse events and low specificity for target lymphocytes.”

We have reviewed references cited in the manuscript and have deleted the following reference 8 because this study does not use oligonucleotide delivery techniques as described in the manuscript (page 5, line 65):

“8. Freeley M, et al. RNAi Screening with Self-Delivering, Synthetic siRNAs for Identification of Genes That Regulate Primary Human T Cell Migration. *J Biomol Screen* **20**, 943-956 (2015).”

We have added new reference (Zhao Y, et al. *Mol Ther* **13**, 151-159 (2006)) which reports high-efficacy RNA transfection to T cells by electroporation (page 5, line 60) as reference 5.

Because pharmacological inhibition studies in the following reference number 26 used not only chlorpromazine but also amiloride, we have added the paper (Chang CC, et al. *Mol Ther Methods Clin Dev* **1**, 14058 (2014)) for the reference of amiloride (page 11, line 178).

We have added the following phrase (page 15, line 251) and the new two papers (Song E, et al. *Nat Biotechnol* **23**, 709-717 (2005) and Peer D, et al. *Proc Natl Acad Sci U S A* **104**, 4095-4100 (2007)) as reference number 50 and 51 respectively:

“antibody fragment fusion protein-conjugates ^{50, 51, 52}”

Comment 9: *HDOs have been conjugated with α -tocopherol to facilitate delivery to lymphocytes via binding to serum lipoproteins in blood and distribution via the α -tocopherol transport pathway. Since α -tocopherol preferentially binds to serum lipoproteins, will this approach result in relatively low HDO delivery efficiency to spleen and other lymphoid organs? Moreover, α -tocopherol is physiologically essential to enhance proliferation and form effective immune synapses by lymphocytes, opening the possibility of potential side effects to HDO- α -tocopherol.*

Response: The previous study (reference 12) showed that unconjugated HDO is comparable in gene silencing efficacy to the parent ASO, and α -tocopherol conjugation to HDO facilitates the binding to serum lipoprotein such as HDL and LDL, resulting in a more specific and efficient delivery along with physiological pathway of α -tocopherol. ASO and HDO itself are small enough to be rapidly filtered out of blood by the kidney and excreted in urine, so allowing HDO to bind to serum proteins maintains HDO in circulation long enough to distribute broadly to tissues, or passive targeting as shown Fig. 3C and Response Fig. 2.

Fig. 3C

(C) Time course of oligonucleotide concentration in plasma and peripheral blood lymphocytes after injection of 20 mg/kg Alexa Fluor 647 (AF647)-labeled Toc-HDO or ASO. Oligonucleotide concentration was calculated by fluorescence intensity.

Response Fig. 2

Time course of oligonucleotide concentration in mouse spleen and lymph node (inguinal and axillary) after injection of 20 mg/kg Alexa Fluor 647-labeled Toc-HDO or ASO over 12 h. Oligonucleotide concentration was calculated by fluorescence intensity. *p* values were calculated using Student's two-tailed *t*-test (*N* = 3 or 4), * *p*<0.05, ** *p*<0.01, *** *p*<0.001).

We attribute highly efficient cellular uptake of Toc-HDO shown in Fig. 3 to high retention in blood, escape from hepatic and renal excretion, and characteristic cellular internalization pathways by α -tocopherol conjugation. As we did not mention that in our previous manuscript, we have added the following text (page 16, line 272):

“Conjugation with α -tocopherol allows HDO to bind to serum lipoprotein such as HDL and LDL, resulting in an efficient delivery along with physiological pathway of α -tocopherol¹². Unconjugated

HDO as well as ASO is small enough to be rapidly filtered out of blood by the kidney and excreted in urine, so binding to serum proteins enhanced by ligand conjugation maintains HDO in circulation long enough to distribute broadly to peripheral tissues. Efficient cellular uptake of Toc-HDO was attributed to multiple roles of α -tocopherol conjugation, including highly retention in blood and escape from renal excretion with the passive targeting effect and efficient internalization through multiple cellular uptake mechanisms.”

The previous studies about α -tocopherol in immunity focused on the beneficial effects on immune responses, such as increased proliferation and cytokine production in response to stimulation or infections. Limited information is available on other immune adverse effects or the molecular mechanisms of immunomodulatory effects by α -tocopherol supplementation. There is general agreement that reactive oxygen species (ROS) contribute to T cell dysfunction by damaging the lipid moieties of membranes as well as enzymatic and structural proteins (*Exp Gerontol* 2007;42:852–858.). Highly lipophilic antioxidant activity of α -tocopherol can neutralize ROS-mediated damage of membrane lipids or adapter proteins/kinase, or prevent the propagation of polyunsaturated fatty acid peroxidation. In this regard, covalent conjugation of α -tocopherol to HDO blocks the hydroxyl moiety of chroman ring where is the antioxidant active site of α -tocopherol, then it is unlikely to cause side effects due to its extreme antioxidant action. On the other hand, α -tocopherol has antioxidant-independent effects, such as intracellularly modulating gene expression and enzyme activities (*J Immunol* 2006;177:6052–6061., *Free radic Biol Med* 2009;46:543–554., *Redox Biol* 2014;2:495–503.). The non-antioxidant regulatory effects can influence the activities of several enzymes involved in signal transduction pathway, for example, to inhibit protein kinase C and phospholipase A2 activities or stimulate protein tyrosine phosphatase (*Free Radic Res* 2001;35:843–856., *J Biol Chem* 1991;266:6188–6194.). In the case of phospholipase A2, structural evidence exists of a direct physical association with α -tocopherol (*J Mol Biol* 2002;320:215–222.). The precise biological functions and molecular mechanisms of α -tocopherol’s non-antioxidant activities remains a matter of debate. Although we detected no effects of Toc-HDO administration on inflammatory cytokines production in wild-type and experimental autoimmune encephalitis model in vivo as shown in our present study, as the reviewer has pointed out, the possibility of unexpected side effects caused by a delivery ligand or HDO itself is undeniable. We have added the following text (page 16, line 272):

“However, we have to pay attention to the possibility of unexpected adverse effects by α -tocopherol conjugation.”

Reviewer #3:

We wish to express our appreciation to the reviewer for his or her insightful comments, which have helped us significantly improve the paper.

Comment 1: *The authors compared Toc-HDOs and the parent gapmer ASOs. Have the authors looked at gapmer ASOs conjugated with α -tocopherol (Toc)?*

Response: The reviewer's comment is reasonable. We evaluated the tocopherol-conjugated ASO (Toc-ASO) in primary T cells. Toc-ASO demonstrated mild gene silencing effects (about 35% knockdown) at a high dose in primary T cells. However, all mouse intravenously-injected with Toc-ASO at corresponding doses to 50 mg/kg of ASO have died within 2 days. Although the cause of death was unknown, these findings suggested direct conjugation of α -tocopherol to the ASO can inhibit the ASO activity in vitro and may lead to lethal complications in vivo. We have added the following text in the manuscript (page 7, line 96):

“First, we evaluate gene silencing effect of ASO, Toc-HDO, and directly-conjugated α -tocopherol to 5'-end of ASO (Toc-ASO) targeting Malat1 in primary T cells in vitro. Although the conventional ASO induced target gene reduction, Toc-HDO induced more efficient gene knockdown. The direct conjugation of α -tocopherol drastically decreased the gene silencing effect even at high doses (Fig. S1C).”

Fig. S1C

(C) Quantitative RT-PCR analyses of *Malat1* RNA levels in mouse primary T cells, which were incubated with *Malat1*-targeting ASO, Toc-HDO, or Toc-ASO at the indicated concentrations for 24 h without any transfection reagents.

And we have added the following text (page 7, line 102):

“We also evaluated the in vivo gene silencing efficacy of Toc-ASO, but all mouse died within 2 days after intravenous injection of Toc-ASO at corresponding doses to 50 mg/kg of the ASO, indicating high toxicity with direct conjugation of α -tocopherol to ASO in vivo. These results suggest that systemic administration of ASO has a mild gene silencing effect on lymphocytes. Direct conjugation of α -tocopherol to ASO prevents the ASO activity and leads to a fatal outcome in vivo, whereas Toc-HDO reduces target gene expression more efficiently than ASO in lymphocytes in vivo, especially peripheral blood lymphocytes.”

Comment 2: For the screening of sequences in gapmers targeting *ITGA4*, *MALAT1* and *DMPK* in vitro, have the authors used negative control oligonucleotides (mismatch and scrambled controls) to validate the on-target effects? For example, in Figure S1A, using WT as negative control is not sufficient.

Response: The reviewer’s comment is reasonable. We have designed two *Itga4*-targeting scrambled controls, estimated the *Itga4* gene expressions by these controls in mouse primary T cells and added these data in Fig. S1A (red circle). Furthermore, we have tested *Malat1*-targeting 3bp mismatched and scrambled controls as previously reported (*Nature Biotechnology* in press) in primary T cells and added in Fig. S1B (red circle). In both target genes, each scrambled and mismatched controls did not induce gene knockdown in vitro. We have added the following phrase (page 7, line 94):

Fig. S1A and B

(A) Screening ASO sequences that efficiently reduce mouse *Itga4* mRNA expression in primary T cells measured by quantitative RT-PCR analyses 24 h after treatment with 500 nM ASO without any

transfection reagents. The ASO 20mer sequence was reported to knockdown *Iga4* mRNA expression in a mouse disease model. (C) Target *Malat1* RNA levels measured by RT-PCR in mouse primary T cells treated with 1 μ M of the indicated *Malat1*-targeting oligonucleotides for 24 h without any transfection reagents.

“Specific ASO sequences for *Malat1* and *Dmpk* have high specificity, efficacy, biological stability, and safety in previous studies^{19,20,21} (Fig. S1B)”

We have added each of scramble and mismatch sequences in materials and methods section and Table S1. (page 20, line 324):

“*Iga4*-targeting scramble ASO sequence: 5'-G[^]C[^]G[^]a[^]c[^]g[^]c[^]g[^]t[^]g[^]a[^]c[^]T[^]C[^]T[^]-3' and 5'-G[^]C[^]G[^]g[^]c[^]g[^]a[^]t[^]c[^]t[^]a[^]t[^]g[^]C[^]T[^]C[^]-3'. *Dmpk* mRNA, *Malat1* RNA, and *Stat3* mRNA targeting sequences: 5'-A[^]C[^]A[^]a[^]t[^]a[^]a[^]a[^]t[^]a[^]c[^]g[^]A[^]G[^]G[^]-3', 5'-C[^]T[^]A[^]g[^]t[^]t[^]c[^]a[^]c[^]t[^]g[^]a[^]a[^]T[^]G[^]C[^]-3', and 5'-C[^]T[^]A[^]t[^]t[^]g[^]a[^]t[^]g[^]t[^]c[^]A[^]G[^]C[^], respectively. *Malat1*-targeting scramble ASO sequence: 5'-A[^]C[^]G[^]t[^]g[^]a[^]t[^]c[^]g[^]c[^]t[^]t[^]A[^]T[^]A[^]-3'. *Malat1*-targeting mismatch ASO sequence: 5'-C[^]T[^]G[^]g[^]t[^]g[^]c[^]a[^]g[^]t[^]g[^]a[^]a[^]T[^]G[^]C[^]-3'.”

Comment 3: *In line 195 and 298, the authors mentioned that besides macropinocytosis and clathrin-independent dynamin-dependent pathways, Toc-HDO is also uniquely uptaken by a clathrin- and dynamin-independent pathway.*

1) *This statement is based on Fig 4D. However, the authors have not tested concentration larger than 100 μ M dynasore. Will higher concentration reverse the inhibition of MALAT1 by Toc-HDO?*

Response: The reviewer’s comment is reasonable. We confirmed that 200 and 300 μ M dynasore treatment can inhibit Toc-HDO activity, but simultaneously cell viability decreased above 100 μ M concentration of dynasore. Percent of cell viability is about 74.9 \pm 4.67% at 100 μ M, 49.6 \pm 4.24% at 200 μ M, and 16.6 \pm 1.23% at 300 μ M of dynasore shown in Response Figure 3. Higher concentration of dynasore may inhibit endocytosis of substances that are essential for lymphocyte survival or have other cellular cytotoxic mechanisms. We have added the following text (page 12, line 185):

“The higher concentration of dynasore were found to be cytotoxic (data not shown).”

Response Fig. 3

Percent of cell viability of EL4 cells measured by flow cytometry analysis using fixable viability dyes staining after 1 h treatment with indicated dose of dynasore.

2) *The authors should list the IC50 of the five specific endocytosis pathway inhibitors to rationalize the different concentration range used in Fig 4 and Fig S2.*

Response: We thank the review for this pertinent comment. In accordance with the reviewer's comment, we have added the supplementary table that lists the IC50 of the five endocytosis pathway inhibitors used in this study, and cited new 6 papers as new references number 25–26, 30, 32–34.

Table S3. Concentrations of endocytosis pathway inhibitors used in this study

Inhibitors	Concentrations used	IC50
Amiloride	1, 3 mM	200–500 µM ^{25,26}
Chlorpromazine	5, 10, 30 µM	17.4 µM ³⁰
Cytochalasin D	10 µM	1.4 µM ³⁴
Dynasore	10, 50, 100 µM	15 µM ³¹
Filipin	1, 2, 3 µM	0.4–2 µM ^{32,33}

The previous manuscript showed Toc-HDO and ASO uptake inhibition in the range of 1–10 µM chlorpromazine, but these ranges were unconvincing enough as IC50 of chlorpromazine reported in the previous studies were 17.4 µM. We have added the gene silencing efficacy of Toc-HDO and ASO in the presence of 30 µM chlorpromazine (Fig. 4E). We also have added the gene knockdown by Toc-HDO and ASO with 20 µM cytochalasin D (Fig. S3B). However, these additional results did not influence our findings about Toc-HDO cellular uptake pathway mechanisms.

Fig. 4E

(E) Quantitative RT-PCR analyses of *Malat1* RNA levels in EL4 cells treated with chlorpromazine, followed by treatment with Toc-HDO or ASO. EL4 cells were incubated with chlorpromazine for 1 h and then treated with 500 nM Toc-HDO or ASO for 4 h, followed by 20 h culture. Data are normalized to *Gapdh* mRNA levels and represent at least two independent experiments. *p* values were calculated using one-way ANOVA with Holm's post-test ($N = 3$, $*p < 0.05$, $**p < 0.01$, $***p < 0.001$).

Fig. S3B

(B) Effects of phagocytosis inhibitor on gene silencing of Toc-HDO and ASO. Target *Malat1* RNA levels in the EL4 cells treated with cytochalasin D, followed by treatment with Toc-HDO or ASO. EL4 cells were incubated with 10 or 20 µM cytochalasin D for 1 h and treated with 500 nM Toc-HDO or ASO, washed, and cultured for another 20 h before RNA isolation.

We have changed to the following text. (page24, line 392)

“Inhibitors were used at the following concentrations: amiloride: 1–3 mM; chlorpromazine: 5–30 µM; dynasore: 10–100 µM; filipin III, 1–3 µM; and cytochalasin D, 10–20 µM (all from Sigma-Aldrich).”

Comment 4: In Fig 2D and Fig S1C, the authors used FACS to quantify the inhibition of ITGA4 protein in sorted T cells and B cells. Is western blot feasible in this protocol? Having gel image as primary data could validate the gene silencing in both mRNA and protein levels.

Response: In accordance with the reviewer's comment, we analyzed ITGA4 protein expression by western blot. The samples were splenic lymphocytes from wild-type mice intravenously-injected with *Itga4*-targeting Toc-HDO or ASO at corresponding doses to 50 mg/kg of the ASO. Quantitative analyses by Fiji image processing demonstrated that Toc-HDO induces significant reduction of ITGA4 protein expression (Fig. 2E). We have changed to the following text (page 8, line 112):

Fig. 2E

(E) ITGA4 protein expression determined by western blot in splenic lymphocytes 5 days after intravenous administration of 50 mg/kg Toc-HDO, ASO, or PBS alone (N = 4).

“Similar data were obtained from splenic lymphocytes by western blot and flow cytometry analysis (Fig. 2E and S2C).”

Furthermore, based on the comment 3 of the reviewer #2, we investigated the protein expression reduction by Toc-HDO targeting human *Stat3* in Jurkat cells by western blot. Toc-HDO can reduce the STAT3 expression at both mRNA and protein levels (Fig. S5B).

Fig. S5B

(B) STAT3 protein expression determined by western blot in Jurkat cells after treatment with *Stat3*-targeting Toc-HDO or ASO for 24 h. Quantitative RT-PCR data and band intensity shown are relative to *Gapdh* mRNA and GAPDH protein levels, respectively. Data are expressed as mean values \pm s.e.m. and are represented two independent experiments. *p* values were calculated using one-way ANOVA with Holm's post-test ($*p < 0.05$, $**p < 0.01$, $***p < 0.001$).

We have further added antibodies for ITGA4 and STAT3 in Table S2 (Antibodies used in this study) and the following text in materials and methods section (page 26, line 428):

“Western blot analysis

Whole cell extracts were isolated using RIPA lysis buffer (50 mM Tris-HCl pH 8.0, 150 mM NaCl, 1% Nonidet P-40, 0.5% sodium deoxycholate, 0.1% SDS) supplemented with protease inhibitors (Roche). The proteins were separated on a 15% SDS-PAGE gel and electrophoretically transferred to PVDF membranes (Millipore). Membranes were then incubated with the blocking buffer followed by incubation with the primary antibodies against Integrin α 4 (Cell Signaling Technology), STAT3 (Cell Signaling Technology) or GAPDH (Wako Pure Chemical Industries, Osaka, Japan) and the appropriate secondary antibodies. Protein band intensities were quantified with Fiji image processing software (National Institutes of Health)⁷².”

Comment 5: *In Fig 2A-C and Fig S1B, the selection of tissues is different. The authors need to provide rationale for it.*

1) *In Fig 2A-B, the authors selected lymph nodes for ITGA4, and thymic lymphocytes for MALAT1.*

Response: We showed the representative tissues but that had misleading connotations. We have added thymic lymphocytes for *Iga4*, and lymph nodes lymphocytes for *Malat1* in Fig. 2A and B. We have added the following phrases (page 7, line 100):

“After 72 h, Toc-HDO significantly reduced *Itga4* mRNA expression compared with an equivalent dose of ASO in peripheral blood, splenic, lymph node, and thymic lymphocytes and bone marrow cells (Fig. 2A).”

Fig. 2A and B

(A, B) Target mRNA levels measured by quantitative RT-PCR in mouse lymphocytes from the indicated tissues 72 h after intravenous injection of 50 mg/kg Toc-HDO, ASO, or PBS alone. (A) *Itga4* (B) *Malat1*.

2) In Fig S1B, the authors selected three tissues instead of four which shown in Fig 2A-B.

Response: Since *Dmpk* mRNA expression in peripheral lymphocytes is not detected by qRT-PCR, we did not show that in previous Fig. S1B (New Fig. S2A). However, we did not mention that in our previous manuscript, so we have added the following text (page 7, line 102):

“*Dmpk* mRNA levels in peripheral lymphocytes were not shown because of those low expressions.”

3) In Fig 2A and 2C, is peripheral blood equal to peripheral lymphocytes? It’s a good idea to mention what types of tissues consist of lymphocytes in vivo for general audience.

Response: As the reviewer pointed out, “peripheral blood” in Fig 2A and 2C is equal to “peripheral lymphocytes”. In accordance with the reviewer’s comment, we have changed “peripheral blood” in Fig. 2A and 2C to “peripheral lymphocytes” and added the following text for general audience. (page 4, line 42):

“The two main types of lymphocytes are T cell and B cell. Both originate from stem cells in the bone marrow, but T cells mature in thymus and B cells mature in the bone marrow. The thymus and bone marrow constitute the primary lymphoid tissues that are the sites of lymphocyte generation and maturation, while the secondary lymphoid tissues, including lymph nodes and spleen, are responsible for maintaining mature naïve lymphocyte and initiating an adaptive immune response through antigen presentation. Following maturation, lymphocytes recirculate through the blood and peripheral lymphoid organs where they survey for invading pathogens.”

Comment 6: *In line 731, Fig 4 legend, the concentration should be 500 nM instead of 500 uM.*

Response: We would like to thank the reviewer for pointing out the error. We have changed “500 μ M” to “500 nM” in page 37, line 736.

Thank you again for your comments on our paper. We trust that the revised manuscript is suitable for publication.

Reviewers' Comments:

Reviewer #1:

Remarks to the Author:

The authors have satisfactorily addressed all comments. They have performed additional experiments that are included in the revised version of the manuscript.

Reviewer #2:

Remarks to the Author:

The authors have satisfactorily addressed this reviewer's concerns in the revised ms.

TOKYO MEDICAL AND DENTAL UNIVERSITY

Response to the Reviewer' comments

Reviewer #1:

The authors have satisfactorily addressed all comments. They have performed additional experiments that are included in the revised version of the manuscript.

Response: We wish to express our appreciation to the Reviewer for his or her insightful comments, which have helped us significantly improve our paper.

Reviewer #2:

The authors have satisfactorily addressed this reviewer's concerns in the revised ms.

Response: We wish to express our appreciation to the Reviewer for his or her insightful comments, which have helped us significantly improve our paper.